# A Comprehensive Review on *Cannabis sativa* Ethnobotany, Phytochemistry, Molecular Docking and Biological Activities

**DOI:** 10.3390/plants12061245

**Published:** 2023-03-09

**Authors:** Sohaib Hourfane, Hicham Mechqoq, Abdellah Yassine Bekkali, João Miguel Rocha, Noureddine El Aouad

**Affiliations:** 1Research Team on Natural Products Chemistry and Smart Technology (NPC-ST), Polydisciplinary Faculty of Larache, Route de Rabat, Abdelmalek Essaadi University, Tetouan 92000, Morocco; 2LEPABE—Laboratory for Process Engineering, Environment, Biotechnology and Energy, Faculty of Engineering, University of Porto, Rua Dr. Roberto Frias, s/n, 4200-465 Porto, Portugal; 3ALiCE—Associate Laboratory in Chemical Engineering, Faculty of Engineering, University of Porto, Rua Dr. Roberto Frias, s/n, 4200-465 Porto, Portugal

**Keywords:** *Cannabis sativa* L., ethnobotany, chemical composition, biological activities, medicine, in silico, molecular docking

## Abstract

For more than a century, *Cannabis* was considered a narcotic and has been banned by lawmakers all over the world. In recent years, interest in this plant has increased due to its therapeutic potential, in addition to a very interesting chemical composition, characterized by the presence of an atypical family of molecules known as phytocannabinoids. With this emerging interest, it is very important to take stock of what research has been conducted so far on the chemistry and biology of *Cannabis sativa*. The aim of this review is to describe the traditional uses, chemical composition and biological activities of different parts of this plant, as well as the molecular docking studies. Information was collected from electronic databases, namely SciFinder, ScienceDirect, PubMed and Web of Science. *Cannabis* is mainly popular for its recreational use, but it is also traditionally used as remedy for the treatment of several diseases, including diabetes, digestive, circulatory, genital, nervous, urinary, skin and respiratory diseases. These biological proprieties are mainly due to the presence of bioactive metabolites represented by more than 550 different molecules. Molecular docking simulations proved the presence of affinities between *Cannabis* compounds and several enzymes responsible for anti-inflammatory, antidiabetic, antiepileptic and anticancer activities. Several biological activities have been evaluated on the metabolites of *Cannabis sativa*, and these works have shown the presence of antioxidant, antibacterial, anticoagulant, antifungal, anti-aflatoxigenic, insecticidal, anti-inflammatory, anticancer, neuroprotective and dermocosmetic activities. This paper presents the up-to-date reported investigations and opens many reflections and further research perspectives.

## 1. Introduction

*Cannabis sativa* L. is an herbaceous plant belonging to the Cannabaceae family. This plant species has many vernacular names and is known by many people as marijuana and hemp. Despite being native to Central Asia, this plant’s capacity of adaption to different climates lead to its spread all over the world [1]. The *Cannabis* genus is composed of a single specie named “*sativa*”, which regroup several subspecies or varieties including *Cannabis sativa* ssp. *sativa*, *Cannabis sativa* ssp. *indica*, *Cannabis sativa* ssp. *ruderalis* and *Cannabis sativa* ssp. *afghanica*. However, there is still controversy among the scientific community about the sub-classification of *Cannabis* species and varieties [2,3,4]. *Cannabis sativa* L. is one of the plants that have been used by humankind since antiquity, and many historians reported the different uses of this plant around the world [5,6,7,8]. The historical records show that this plant has been used as a source of fiber, food, oil, as well as for recreational and religious purposes. Additionally, several other uses have been developed through the centuries, such as livestock feed, skin and hair care [9]. Furthermore, many ethnobotanical surveys highlighted the therapeutic use of *Cannabis sativa* L. for the treatment of chronic pain, depression and inflammation. These activities have been justified by the original chemical composition, viz. Cannabis contains a large number of bioactive compounds with an estimation of more than 550 molecules [10]. Those compounds belong to the cannabinoid, terpenoid, stilbenoid, lignanamide, carotenoid, flavonoid and alkaloid classes [11]. The most notable compound of Cannabis remains the cannabinoids [12,13], a class of terpenolic compounds mainly found in the trichome cavity of female flowers [14,15]. Nowadays, *Cannabis sativa* L. is experiencing a renewed interest in many research fields, including microbiology and oncology [12,13]. In fact, the chemical diversity of cannabinoids proved to be very useful for targeting microorganisms such as bacteria, fungi, viruses, as well as cell components such as proteins and genes [16]. Additionally, their natural origin and low toxicity make them perfect candidates to treat hard-to-treat diseases, by solving therapeutic problems such as the resistance to antibiotics and, in the case of cancer treatment, the toxicity induced by ingestion and metabolization [17,18,19].

This review aims to present *Cannabis sativa* subspecies classification, description of the plant aspect and botany as well as a brief history and geographic distribution. This manuscript aims also to report and discuss the traditional uses as to both the preparation and administration modes of every part of the plant. It also aims to take into account the chemical composition of each part with the classification of the identified metabolites and their quantification, and the biological activities of Cannabis extracts and purified compounds. Finally, this review also includes the molecular docking studies of secondary metabolites previously identified in different parts of *Cannabis sativa* L.

## 2. Generalities about *Cannabis sativa* L.

### 2.1. Plant Nomenclature and Synonyms

*Carolus Linnæus,* also known as Carl von Linné (1707–1778), was the first person to frame principles for classification of living organisms into classes and sub-classes. His aim was to create a uniform international system for the identification of any living organism according to its morphological features. In this system, every organism is identified by his genera and specie names known as “binomial nomenclature”. In 1753, Carl von Linné mentioned the word Cannabis for the first time. This word comes from the Latin *canna* that means “reed” and *bis* that means “twice”, which means literally “reed with two sexes” [20]. Prior to the Linnæus nomenclature, Cannabis was widely used by different civilizations that gave it different names known as vernacular names [21,22,23]. At present, there are many local or vernacular names and various synonyms to name Cannabis. It is also known as hashish, marijuana, weed, Acapulco gold, ace, bat, bhang, log, hemp, Indian hemp, Colombian, doobie, dope (Cannabis), ganja, hydro, Jamaican, jive (sticks), joint, Maui wowie, Mexican, Panama gold, Panama red, pot, firecracker, ragweed, reefer, sativa, sinsemilla of California, spliff, Thai stick, etc. Those names and designations stay different depending on the region, country and culture. *Cannabis sativa* belongs to the Cannabaceae family, which includes 12 genera and 102 species, and with some species of economic importance, such as *Humulus lupulus* L. and *Pteroceltis tatarinowii* [24]. There are conflicting botanical classifications of *Cannabis sativa*, and the taxonomic classification of this plant has been the subject of divergences and debates. It is commonly accepted and recommended that *Cannabis sativa* is a single species [25], with four subspecies, namely *indica*, *ruderalis, sativa* and *afghanica* [2,3,4,26]. However, the classification criteria used for the differentiation of *Cannabis sativa* subspecies are often not very clear, since the chemical and morphological characteristics appear to vary according to the plant environment and pedology. In a study reported by Pacifico, et al. [27], the authors showed that the tetrahydrocannabinol (THC) content of a *Cannabis sativa* single species depends on the growing climate of the plant. In most cases, it is recommended to apply the name *Cannabis sativa* to all Cannabis plants encountered, since they all belong to the same species, and there is no agreement on the plant taxonomy [25].

### 2.2. Description and Botanical Aspect

*Cannabis sativa* L. is an annual, usually dioecious plant belonging to the *Cannabaceae* family [28]. It is now considered as the only species of the botanical genus Cannabis but divided into several phenotypes that can be described as subspecies or varieties [29]. *Cannabis sativa* has the particularity of being a fast-growing plant with a fluted stem that can reach 1 to 4 m with a diameter ranging between 1 and 3 cm (Figure 1a) [30]. The variation of height and diameter depends on the sub-species, environment, soil and climatic conditions [31,32]. The seeds are smooth, greyish ovoid or spherical in shape, 2.5 to 3.5 mm long and 2.5 to 3 mm in diameter (Figure 1c). Each seed contains two cotyledons rich in reserves (protein and oil), with an albumen considered particularly small compared to other plant species [33].

This plant is also characterized by long, fine flowers (Figure 1b). It has glandular hairs that make it fragrant and sticky [34,35]. At post-germination, young male and female plants cannot be distinguished. It is only during the last phase of growth, when flowers start appearing, that sex determination becomes possible [24,36]. The female flowers have no petals and consist of two long white, yellow or pink stigmas. Their calyx (less than 3–6 mm) envelops the ovary containing a single ovule. The female flowers appear in pairs in the axils of small leaves named bracts, these bracts contain numerous glandular trichomes where cannabinoids, mainly THC, accumulate [34,37,38]. On the other hand, the male flowers have five sepals of approximately 5 mm length, with yellow, white or green color [33,39]. The male plants develop small pollen sacs that serve to fertilize the female plants with hairy, resinous stigmas [34,36,40]. The Cannabis leaves are stipulate and opposite, with palmate (five to seven unequal), elongated and spiny segments with toothed margins (Figure 1d). Towards the top of the axis, the leaves become alternate and are inserted on the stem in an opposite arrangement every 10–30 cm [39]. These plants have cystolithic, tectorial and resin-secreting hairs; the latter have a voluminous base ending in a cluster of several cells, with each one secreting resin [39]. The root is taproot with a length of up to 30 cm, but the lateral roots reach 20 to 100 cm. In addition, in peaty soils, the lateral roots are more strongly developed, and the main root grows to a depth of 10–20 cm [41]. The growth rate of the root system is quite slow in the initial stages of vegetation, in contrast to the aerial part of the Cannabis plant, which grows intensively and rapidly [41].

### 2.3. Geographic Distribution and History

In nature, Cannabis is an annual flowering plant. This means that it completes its life cycle, from germination to seed production, in one year [42]. Cannabis can grow in a vast majority of climates (Figure 2). From its region of origin, it appreciates calcareous and nitrogenous soils with a neutral or slightly acidic pH [43,44].

This species originates from equatorial and subtropical regions, mainly from central Asia [1], where two places seem to be its cradle: the foothills of the Himalayas and the plains of the Pamir (a high mountain range centered in eastern Tajikistan with extensions into Afghanistan, the Republic of China and Kyrgyzstan) [45]. However, this plant has a wide geographical distribution growing up in Canada, United States of America, Europe and Africa. Cannabis is an ancient plant but the craze it has generated over (at least) the last century has greatly changed its face and even the face of the world. It is probably the first plant domesticated by humankind [46]. Many historical reports prove that this plant had been cultivated worldwide for thousands of years. The oldest documented evidence of Cannabis cultivation is a 26,900 B.C. hemp rope found in the Czech Republic [47]. Some of the earliest known prolific uses of Cannabis began in China around 10,000 B.C., where Cannabis was used to make clothing, rope and paper [48]. Further traces were reportedly found at the Neolithic site of Xianrendong on Chinese ceramics dating back to 8000 B.C. and decorated with hemp braided fibers. Between 8000 and 300 B.C., Cannabis was also cultivated in Japan and employed to make cloth fiber and paper [49,50]. However, the earliest reference of Cannabis psychotropic use goes back to 2700 B.C. It has been mentioned in the Chinese pharmacopoeia of the Emperor Chen Nong, where it is recommended as a sedative and remedy for insanity. Cannabis was also mentioned on the Ebers Papyrus of pharaonic Egypt back to 1550 B.C. as remedy for vaginal inflammations [51]. Yet, it was mentioned in Greek medicine, in the writings of Dioscorides, who underlines the psychotropic properties of the plant and already Galen fears that “it hurts the brain when we take too much” [52,53]. In India, it was one of the five magical plants used in religious rituals in the form of fumigation. In fact, around 1300 B.C., the stimulating and euphoric powers of bhanga (hemp in Sanskrit) were praised by the Indo-Aryans in one of the four holy books, the Atharva Veda [54]. Back to the European Continent, and around 700 B.C. in Marseille (France), Cannabis was used for rope manufacturing. The name Cannebiere (important avenue of the city) testifies of the importance of Cannabis at that time [35]. Jamestown settlers introduced Cannabis to colonial America in the early 1600s for the manufacture of rope, paper and other fiber products. This plant was so important that American presidents George Washington and Thomas Jefferson grew Cannabis [38]. The question of when and how Cannabis originated in the new world is still very controversial indeed. Cannabis was discovered in native American civilizations prior to Columbus’ arrival [55]. William Henry Holmes’ 1896 report “prehistoric textile art of the Eastern United States” indicated that Cannabis originated with native American tribes of the Great Lakes and Mississippi valley [56]. Cannabis products from pre-Columbian indigenous civilizations have also been found in Virginia [57]. Cannabis was an important crop in the United States until 1937, when the Marihuana Tax Act all but wiped out the American hemp industry. During World War II, Cannabis experienced a resurgence in the United States of America, as it was widely used to manufacture military items ranging from uniforms to canvas and rope [57]. At present, the most notable development in Cannabis production around the world is the rise of indoor cultivation, particularly in Europe, Australia and North America. This type of cultivation gives rise to a very lucrative trade, which is increasingly a source of profit for local organized crime groups [58].

## 3. Methodology

Relevant information about *Cannabis sativa* L. was collected from various scientific sources including SciFinder, ScienceDirect, PubMed and Web of Science. The targeted databases were probed with “Cannabis sativa”, “botany”, “history”, “ethnobotany”, “traditional use”, “phytochemistry”, “pharmacology”, “bioactivity”, “bioinformatic” and “in-silico prediction” as keywords. Thus, available articles were collected, summarized in tables and analyzed. In addition to that, we report up-to-date studies of *Cannabis sativa* ethnopharmacology, chemical composition, pharmacology and molecular docking simulations; this review aims to give a subjective critique to reported articles and offer perspectives for further investigations on Cannabis phytochemistry and pharmacology.

## 4. Results and Discussion

### 4.1. Traditional Uses of Cannabis sativa L.

*Cannabis sativa* L. has been used in a wide variety of fields and showed a high usability potential with many applications including manufacturing of tools, construction, cosmetics, medication, shelter insulation, papermaking, human nutrition, animal feed, agrofuels, composite materials in association with plastics, etc. [5,6,7,8]. Table 1 summarizes the parts of the plant and their traditional uses.

The analysis of collected data shows that seeds and leaves are the most used parts for medication. Many studies reported the use of Cannabis seed as food. It is used for the production of pasta, gluten-free flour characterized by a nutty taste, beer and oil [72,73]. In addition to their use for human nutrition [59], the seeds are also used to treat nausea, vomiting, stimulate the appetite of AIDS patients, cancer and hepatitis C. It is also applied as a muscle relaxant, for weight control, lung capacity enhancer and as an analgesic, anxiolytic, antiepileptic, antiemetic and against neurological pain [60]. These seeds have also a cosmetic use, mainly for hair fortification as a hair serum by external application of seed powder [61]. Beyond their potential value as medicine or food, *Cannabis sativa* seeds have recently been used to treat contaminated groundwater, since hemp seed protein powder proved to be more effective than other plant protein sources for chelating perfluoroalkyl and polyfluoroalkyl substances, known as “eternal chemicals” [74]. Those Cannabis proteins proved to be very useful for the treatment of salt-contaminated soils as well [75].

The leaves are externally used as poultice to treat eczema [61] and subcutaneous tissue disorders [62]. They are also orally consumed by local people for the treatment of central nervous system (CNS) disorders such as schizophrenia, gout, arthritic pain, bloating, coughing and mucus [63,64]. Cannabis leaves are among the most iconic symbols of modern stoner culture; their shape is frequently associated with the recreative use of this plant. Several studies described the traditional uses of Cannabis leaves, and those applications include the treatment of a wide range of health problems such as hypertension, rheumatoid arthritis, itching, cancer, snake and scorpion poisoning [66], as well as gastric and circulatory system disorders [62]. These leaves have also been described as strong analgesics, sedatives and narcotics [76]. Other studies described the use of *Cannabis sativa* stem fibers as firewood [62], for construction, tools, clothes, paper and rope manufacturing [5,59]. These fibers are obtained by a process called defibration, which can be briefly described as a stem beating and grinding. During this process, two co-products are obtained, namely chenevotte and Cannabis dust. It is important to highlight that the quality of the fiber decreases with the maturity of the plant, since the fibers become harder and coarser. In addition to the aerial parts of Cannabis plant, the root parts are also used for medication. They are used in particular for the treatment of joint pain, skin burns, inflammation, vermin and erysipelas infection [70]. These roots are also orally used in the form of juice to relieve issues stemming from childbirth, postpartum and hemorrhage [70]. Among the most cited uses of *Cannabis sativa*, the psychoactive remains the most present. The leaves and inflorescences have been consumed as a narcotic in different forms and have been prepared using different methods; for instance, the leaves are smoked or prepared, the inflorescences or resin are processed into charas or attar, hashish, ganja and plant powder, whereas leaves, inflorescences and shoots are used to prepare drinks (e.g., bhang, thandai, tandai, etc.) [5,77]. Moreover, Klauke, et al. [62] reported the religious use of Cannabis drinks. This plant preparation, referred to as traditional bhang drink, is highly consumed during Indian festivals such as Shivaratri and Holi [78]. Some ethnobotanical surveys described the used of the whole Cannabis plant. The aerial parts are mostly used for the treatment of mental disorders and nervous-system-related conditions. However, the most common use of those parts is for the treatment of gastric disorders, diabetes, scarring and asthma [71]. Conversely, some studies reported the appetite-stimulating, antidysentery and antidiarrhea effects of Cannabis inflorescences, but omitted the mention of preparation and administration modes [62]. The analysis of ethnobotanical findings shows that *Cannabis sativa* is a plant that was integrally exploited by local populations; some traditional uses are common to many countries whereas others are specific to some cultures. One can cite the psychotropic, medicinal and cosmetic purposes found all over the world, in contrast with the religious uses exclusively reported in Asian and Latin American countries. The previously reported activities and proprieties are mainly due to the presence of metabolites with interesting chemical structures; the ethnobotanical uses of Cannabis attracted phytochemists to investigate its chemical composition. The first compound to be identified and isolated from Cannabis was cannabinol at the end of the 19th century [79].

### 4.2. Chemical Composition of Cannabis sativa L.

Numerous studies have shown the importance of Cannabis secondary metabolites as well as their roles. This plant offers a rich reservoir of bioactive molecules that can be used for the production of pharmaceutical, nutraceutical and cosmetic products. Table 2 regroups the chemical composition of *Cannabis sativa* seeds, flowers, leaves and resin.

The chemical investigations conducted in different *Cannabis sativa* plant parts shows that terpenes, polyphenols and cannabinoids are the main represented secondary metabolites. Terpenes are represented by more than 100 molecules identified in the flowers, roots and leaves, as well as in the secretory glandular hairs considered as the main production site [11,98]. Furthermore, more than 20 polyphenols have been identified, and they are mainly flavonoids belonging to the flavone and flavonol subclasses [99]. Concerning the cannabinoids, they are among the most represented metabolites of Cannabis despite being represented by less than 20 molecules.

Cannabis seeds contain approximately 40% oil, 30% fibers and 25% proteins [85,100]. Those oils are rich in triacylglycerols (TAGs) represented by 18 different molecules; the predominating tags were LLL and OLLD with respective values of 23 and 19% [100]. Moreover, those oils contain high amounts of polyunsaturated fatty acids, which represent approximately 80% total fatty acids [101]. The fatty acid composition is characterized by the predominance of linoleic acid with range values of 45–60%, followed by oleic acid and palmitic with respective range values of 15–40% and 5–6% [80,81].

Hemp seeds contains also considerable amounts of polyphenols and tocopherols. According to Babiker, et al. [81], the hydroalcoholic extract of Cannabis seeds contained many polyphenols such as gallic acid (12.9 ± 18.3 mg/100 g) and catechin (6.0 ± 5.2 mg/100 g), whereas Moccia, et al. [83] reported the presence of additional polyphenols, namely quercetin-o-glucoside, n-trans-caffeoyltyramine and rutin. Moreover, another sub-class of polyphenols known as cannabisins have been reported on the seed hydroalcoholic maceration by Moccia, et al. [83]. The last authors described the presence of 11 molecules, namely cannabisin A, B, C, D, E, F, G, I, M, N and O, although these compounds have not been quantified. Concerning the tocopherols, four different isomers have been identified: the lead tocopherols are γ- and δ-tocopherols with 426 and 33 mg/kg, respectively [82,84].

Cannabinoids are a group of C21 or C22 terpenolic compounds mainly produced in Cannabis. They have also been reported in other plant species of the genus *Radula* and *Helichrysum* [102]. These cannabinoids are mainly present in leaves and inflorescences. However, Marzorati, et al. [103] reported the presence of some cannabinoids in the hydroalcoholic maceration of seeds, and Stambouli, et al. [104] reported the presence of mainly the cannabinoids tetrahydrocannabinol (THC), cannabidiol (CBD) and cannabigerol (CBG) in seed oil. These results are probably due to contamination of seeds by inflorescence resins, since cannabinoids are not produced or transported to the seeds [105]. Moreover, phytosterols, a group of lipids with a structure similar to cholesterols, mainly found in vegetable oils, have been identified in seed oils. Aiello, et al. [85] and Stambouli, et al. [84] mentioned the presence of this class of metabolites represented by the β-sitosterol, with 65–90%, and campesterol, with 6–17%. Furthermore, Stambouli, et al. [84] described the presence of an additional phytosterol known as δ-5-avenasterol, with a value of 7.8%. The carotenoids are among the less represented compounds in Cannabis seeds. They have been identified in the seed ethanolic extract, whereas lutein and β-carotene have been reported as major sterols, with 2.5 and 0.5 mg/100 g of dry weight, respectively [82].

The composition in hemp seeds of fatty acids, polyphenols, phytosterols, proteins and fibers, and precisely the presence of insoluble fibers in addition to a wide variety of minerals represented by phosphorus, potassium, magnesium, sulfur and calcium, as well as modest amounts of iron and zinc (an important enzyme cofactor for immunity and food absorption), widely justifies its biological proprieties and importance for human nutrition [106,107].

The Cannabis leaves contain terpenes, polyphenols, cannabinoids and alkaloids. The leaf essential oils are characterized by the presence of (e)-caryophyllene (28.3 ± 4.1%), α-humulene (9.3 ± 1.1%), β-selinene (4.7 ± 0.9%), caryophyllene oxide (4.3 ± 0.9%), α-selinene (3.1± 0.6%) and α-trans-bergamotene (2.7 ± 0.5%) [86]. These volatile terpenes are generally found in photosynthetic plant leaves and plays the role of protection from parasites and water loss [108]. The polyphenols of Cannabis leaves are mainly flavonoids and glycosides with apigenin and luteolin. These flavonoids represent 4 mg per g of the plant material. Elsewhere, the cannabinoids of Cannabis leaves had been described by Nagy, et al. [86]. The authors reported the presence of cannabidiol (CBD), cannabidivarine (CBDV), tetrahydrocannabinol (THC) and cannabichromene (CBC) with 11, 0.8, 0.7 and 0.5%, respectively. Moreover, Zagórska-Dziok, et al. [87] reported the lead presence of CBDA and CBD, with respective values of 150 and 31 mg/g of dry matter. Similarly to the cannabinoids, the alkaloids protect plants from predators and regulate plant growth [109]. These alkaloids have been described in cannabis leaves; Fasakin, et al. [88] described the presence of some alkaloids, with cannabisativine (410.30 μg/g), cannabimine C (376.12 μg/g) and anhydrocannabisativine (218.11 μg/g) as lead compounds.

The flowers show a chemical composition qualitatively similar to the leaves. They are mainly composed of terpenes, polyphenols and cannabinoids. The Cannabis flower terpenes have been identified in the essential oils obtained by hydrodistillation. According to Nagy, et al. [86], these essential oils are mainly composed of (E)-caryophyllene, α-humulene, β-selinene and α-selinene, with values of 29, 10, 4 and 3%, respectively. Additionally, Fischedick, et al. [110] reported the presence of mono- and sesquiterpenes in Cannabis flower essential oils. This study featured the quantification of monoterpenes and proved that they dominate the chemical composition, with a concentration of 28.3 mg/g of dry weight. These monoterpenes are represented by d-limonene, β-myrcene, α- and β-pinene. Furthermore, sesquiterpenes are represented by β-caryophyllene and α- humulene. These sesquiterpenes represent a concentration ranging between 0.5 and 10.1 mg/g of dry weight [110]. Polyphenols and cannabinoids have also been identified in the methanolic extract of flowers. The main polyphenols of flowers are quercetin di-c-hexoside and luteolin c-hexoside-2″-o-hexoside with 2.55 mg/g and 1.01 mg/g, respectively [86]. On the other hand, the cannabinoids are represented by cannabidiol (CBD), tetrahydrocannabinol (THC), cannabidivarine (CBDV), cannabicitran (CBTC) and cannabichromene (CBC), with respective values of 25, 1.5, 1.5, 1 and 0.2% [86]. The cannabinoidic composition of flowers fluctuates due to several environmental factors (e.g., temperature, soil nutrients, desiccation, insect predation and ultraviolet radiation) and genetic factors (varieties and heredity) [111]. According to Yang, et al. [112], the total THC, CBD and CBG increases significantly as the flowers matures, reaching the highest concentration during 6 to 7 weeks after anthesis.

The inflorescences of Cannabis have also been evaluated for their composition, and similar classes of metabolites have been described. These compounds belong to the same classes previously reported for the leaves and flowers. The essential oils of inflorescences have been reported in two research manuscripts. The first, published by Laznik, et al. [90], reported the presence of transcaryophyllene (38.2 ± 1.7%), nerolidol (12.7 ± 1.2%) and α-pinene (11.8 ± 0.4%), whereas the second, published by Pieracci et al. [91], reported the presence of β-caryophyllene (14.4 ± 0.89%), caryophyllene oxide (7.0 ± 1.06%) and α-humulene (5.3 ± 0.10%) as lead compounds. This variation is probably due to the variation of *Cannabis sativa* subspecies, cultivars and/or geographic, climatic and pedologic parameters. Laznik, et al. [90] also reported the chemical composition of inflorescence methanolic extract and concluded that this extract contained mainly cannabinoids represented by cannabidiolic acid (CBDA) and cannabidiol (CBD) with 9.5 and 8.7%, respectively.

Several studies have also described the composition of other *Cannabis sativa* L. parts, including roots, stem and resin. Sakakibara, et al. [113] and Lesma, et al. [114] reported the presence of phenolic amides and lignanamides in the fruits and roots of Cannabis. These compounds belong to the classes of lignans and polyphenols and are mainly cannabisin (A, B, C, D, E, F and G) and grossamide [99]. Triterpenes have also been found in Cannabis roots in form of friedelin and epifriedelanol [115]. The glandular secretory hairs are the main site of resin production. The latter is a yellow and sticky substance which contains the active principles. Some studies have shown its chemical composition. Stambouli, et al. [97] reported a high concentration of THC with a value higher than 20%. More recently, Elkins, et al. [96] identified and confirmed the high content of cannabinoids including cannabidiol (CBD) with a concentration of 72.12 μg/mL, followed by tetrahydrocannabinol (THC) with 48.02 μg/mL and cannabichromene (CBC) with 4.78 μg/mL. The concentrations of *Cannabis sativa* secondary metabolites depend on tissue type, age, variety, growing conditions (soil nutrition, temperature, humidity, UV radiation or light), harvest time (maturity) and storage conditions [80,111,116,117]. The Cannabis seeds are rich in oils and starch mainly composed of terpenes [118]. The seed oils serve as food for the young plant during the early stages of germination. In fact, the plant embryo needs a source of nutrition prior to its contact with soil and air [119]. Moreover, the leaves, flowers and inflorescences are rich in volatile terpenes, polyphenols and cannabinoids. These metabolites are generally involved in defense against ultraviolet radiation or aggression by pathogens. Unlike polyphenols, cannabinoids are more interesting due to their chemical structures and representativeness of Cannabis genus. The study of chemical composition of Cannabis plant parts is very important for the determination of biological activities. It is possible to predict a biological activity from the exhaustive chemical composition of an extract using bioinformatics tools such as molecular docking. This approach has been widely applied on Cannabis secondary metabolites, as described in the next section.

### 4.3. Molecular Docking Studies of Cannabis sativa L.

Molecular docking is a computational approach aiming to predict potential interactions between one or more ligands and a protein [120]. This approach predicts the optimal spatial conformation and orientation of the ligand within a protein active site, in addition to the determination of the interaction mode and binding affinity represented by a score. Molecular docking is the in silico equivalent of real high-throughput screening in which many molecules are tested against biological targets. The main goal of this approach is to discriminate active and inactive agents, in order to identify new molecules that will serve as a starting point for medicinal chemists [121].

The process of molecular docking can be subdivided into two basic steps, namely docking and scoring [122]. Docking is the step in which all possible spatial interactions between a ligand and a receptor are tested in order to identify the optimal interactions, whereas the binding affinity between the ligand and the receptor are quantified in scoring, and a score is given to the poses recorded after the docking phase.

Currently, there are more than 30 available docking software packages [123]. Most of them are also designed for virtual screening (independent dockings of multiple ligands with a protein). The three most frequently cited docking tools are AutoDock, GOLD and flex; they represent 27, 15 and 11% of the references, respectively [124,125].

Despite being very useful for guiding the selection of bioactive molecules for in vitro testing, the molecular docking simulations remains a prediction and can sometimes give erroneous results. They can be expressed either as a false negative, when an active molecule gives low docking affinity, or a false positive, when a non-active molecule is identified as a strong ligand [126]. However, this approach remains useful as a pre-investigation predictive tool. Concerning the Cannabis plant, several molecular docking studies have been reported on the different classes of metabolites, namely cannabinoids, terpenes, polyphenols, flavonoids, lignanamides, alkaloids, vitamins and proteins. These classes have been probed against a large number of specific enzymes that instigate important roles in different physiological processes (digestion, nerve conduction, hormone synthesis, etc.). Results of molecular docking are expressed in kcal/mol, and the lowest values correspond to higher affinity between a ligand and a protein. The studies reported in the bibliography are summarized in Table 3.

#### 4.3.1. Pesticidal Activity

Cholinesterase is an enzyme that catalyzes the hydrolysis reaction of a choline ester (acetylcholine, butyrylcholine) into choline and acetic acid. Acetylcholine is a well-known excitatory neurotransmitter that causes muscle contraction and stimulates the release of certain hormones. The inhibition of choline esterase causes the disfunction of nerve impulse transmission inducing mortality, and this activity is highly coveted for the elimination of pests and insects. The molecular docking of cannabis secondary metabolites against two types of cholinesterase, namely acetylcholinesterase (ACHE) and butyrylcholinesterase (BCHE), was described by Karimi, et al. [127]. In this article, the authors tested compounds belonging to the cannabinoid, flavonoid, terpene and phytosterol classes. For acetylcholinesterase, cannabioxepane, δ-9-THCA, Δ-8-THC and CBN showed scores lower than −10 kcal/mol, whereas cannabioxepane, CBL, CBN, CBT and Δ-8-THC showed scores lower than −8.5 kcal/mol against butyrylcholinesterase. Another investigation reported by Nasreen, et al. [128] reported the acetylcholinesterase docking with some cannabinoids, and CBD showed the best score with a value of −14.38 kcal/mol. Despite using the same docking software and acetylcholinesterase three-dimensional structure, the CBD score is better than the ones previously reported by Karimi, et al. [127]. This difference of results can be explained by the variation of docking parameters, such as the gridbox dimensions, position and the exhaustiveness.

#### 4.3.2. Antimalarial and Anti-Leishmania Activities

The antimalarial activity of cannabinoids from *Cannabis sativa* was reported by Quan, et al. [129]. Their studied cannabinoids were docked against plasmodium falciparum dihydrofolate reductase-thymidinesynthase to recognize the potential binding affinities of these phytochemicals. Furthermore, the in silico antileishmanial activity of phytochemicals from *Cannabis sativa* has been well reported in the literature. In the studies conducted by Ogungbe, et al. [130], a molecular docking analysis was performed to examine the potential leishmania protein targets of plant-derived antiprotozoal polyphenolic compounds. A total of 352 phenolic phytochemicals—including 10 aurones, 6 cannabinoids, 34 chalcones, 20 chromenes, 52 coumarins, 92 flavonoids, 41 isoflavonoids, 52 lignans, 25 quinones, 8 stilbenoids, 9 xanthones and 3 miscellaneous phenolic compounds—were used in the virtual screening study with 24 leishmania enzymes (52 different protein structures from the protein data bank). Notable target proteins were leishmania dihydroorotate dehydrogenase, n-myristoyl transferase, phosphodiesterase b1, pteridine reductase, methionyl-trna synthetase, tyrosyl-trna synthetase, uridine diphosphate-glucose pyrophosphorylase, nicotinamidase and glycerol-3-phosphate dehydrogenase. The results showed that docked polyphenols can be considered as promising drug leads deserving further investigation.

#### 4.3.3. Antiviral Activity

Despite great advances in medical and pharmaceutical research in recent years, diseases caused by viruses have remained a huge burden on public health like coronavirus, in particular SARS-CoV-2. In silico studies, including molecular docking, have repeatedly proved to be useful in addressing the particular challenges of antiviral drug discovery. A study published by Srivastava et al. [131] showed that cannabidiol may have an good affinity with a COVID-19 protease, and such affinity is represented by a score value of −7.10 kcal/mol. In another study, cannflavin exhibited a better score against an HIV-protease with a score of −9.70 kcal/mol [132].

#### 4.3.4. Anti-Inflammatory Activity

Inflammation is a natural body reaction to injury and infection. It is mainly due to the deployment of immune system cells to the site of the injury or infection. The four symptoms of inflammation are heat, redness, swelling and pain. However, anti-inflammatory drugs are used to combat inflammation regardless of the cause of the inflammation. They are symptomatic treatments, i.e., they do not eliminate the cause of the inflammation but only its consequence and have an analgesic action. In a recent study, a panel of proteins, including the cellular tumor antigen p53, the essential modulator of NF-KB, the tumor necrosis factor (TNF) receptor, the transcription factor p65, NF-KB p105, the NF-k-b α inhibitor, the inhibitor of nuclear factor k-b kinase α subunit and the epidermal growth factor receptor, were identified as a primary target implicated in cannabidiol (CBD) anti-inflammatory activity. This finding was supported by molecular docking, which showed interactions between the major proteins and CBD. In addition, several signaling pathways, including TCF, toll-like receptors, mitogen-activated protein kinases, nuclear factor kappa, activated b-cell light chain activator and nucleotide-binding oligomerization domain receptors, were identified as key regulators in mediating the anti-inflammatory activity of CBD [136].

#### 4.3.5. Anticancer Activity

Several molecular docking studies were interested in the potential anticancer activity of molecules derived from *Cannabis sativa* L. The molecular docking performed on placental aromatase cytochrome p450 was reported by Baroi, et al. [138]. These authors reported that cannabinoids, mainly cannabidiorcol and cannabidivarin, potentially bind with the best binding energies of −9.03 kcal/mol and −8.34 kcal/mol, respectively. In another study, molecular docking calculations were also performed to investigate the binding affinity of cannabinoids in the active site of crystal structure of the DLC1 RhoGAP domain in liver cancer 1. According to the performed calculations, cannabichromene and cannabidiolic acid showed promising results with regard to binding affinity to the target GTPase-activating proteins [137]. These compounds are held within the active site by a variety of non-covalent interactions, in particular hydrogen bonds, involving important amino acids. Similarly, cannabinoids were docked to predict their anti-inflammatory and anticancer activity. The researchers reported that cannabigerol and cannabichromene potentially bind to arachidonate 5-lypoxygenase with a binding energy score of −5.34 kcal/mol and −5.14 kcal/mol, respectively [142]. Cannabis flavonoids were also docked against topoisomerase II α to investigate the binding affinity of flavonoids in the active site of topoisomerase II α. The results showed that docked flavonoids can be considered as promising drug leads, thus deserving further investigation.

#### 4.3.6. Antiepileptic Activity

A study published by Li, et al. [146] aimed to examine the mechanism of action of Cannabis on epilepsy, focusing on key compounds, targets and pathways. The molecular docking simulations were applied to identify the active ingredients and potential targets of Cannabis in the treatment of epilepsy. Topological analysis showed that cannabinoid receptor 1, albumin and glycogen synthase kinase-3 β (cnr1, alb and gsk3b) were the key targets with intense interaction. The results showed that cannabinol methyl ether could be the lead compound on the basis of molecular docking against docked protein targets. Therefore, these studies shed holistic light on the active components of Cannabis, which contributes to the search for lead compounds and the development of new drugs for the treatment of neurological diseases.

#### 4.3.7. Neuroprotective Activity

Neuroprotective agents target the various deleterious mechanisms that occur in cerebral ischemia, with the aim of limiting the extension of the ischemic heart. It has been shown that the cannabinoid substances contained in the *Cannabis sativa* plant have great potential in a wide variety of therapeutic applications. However, its neuroprotective capacity has been the most studied in diseases such as Alzheimer’s disease, Parkinson’s disease, Huntington’s disease, multiple sclerosis and amyotrophic lateral sclerosis [151]. The only example of proteins as infectious agents leading to neurodegenerative disorders was the prion protein. Since then, the characteristic self-seeding mechanism of the prion protein has also been attributed to other proteins associated with neurodegenerative diseases, notably amyloid-β (aβ). Modelling with the aβ monomer and pentamer revealed that cannabinoids interacted with the aβ protein mainly through steric interactions and hydrogen bonds. The results showed that CBG bound with the highest affinity of all the docked cannabinoids. The authors of this study reported that this was mainly due to the presence of a geranyl side chain in the CBG structure, as this side chain is associated with increased lipophilicity and may, therefore, increase the propensity to bind in the hydrophobic groove of the pentamer [152].

#### 4.3.8. Dermocosmetic Activities

Dermocosmetic products are on the borderline between cosmetic products and medicines. Considered as cosmetic products, they are mainly used to ensure photoprotection of the skin, i.e., to limit the effects of its exposure to solar radiation, but also to improve the appearance of dry or aged skin, reduce inflammatory dermatological conditions (acne, couperose, seborrheic and atopic dermatitis, psoriasis, etc.), as well as for the care of nails and hair [153]. Furthermore, tyrosinase is a key enzyme in the process of melanogenesis (the biosynthesis of melanin). The dermocosmetic potential of Cannabis polyphenols has been reported, and the results showed a good affinity with the tyrosine phosphatase-1b with a free energy score of −24.34 kcal/mol [149]. Similarly, Cannabis alkaloids have been shown to possess affinity with tyrosinase with score a value of −3 kcal/mol [154].

### 4.4. Biological Activities of Cannabis sativa L.

Studies with *Cannabis sativa* L. have shown the presence of several biological activities, such as antioxidant, antibacterial, anticoagulant, insecticide, anticancer, anti-aflatoxigenic, antifungal, cytotoxic, anti-elastase, anti-collagenase, anti-acetylcholinesterase, anti-inflammatory, neuroprotective (anti-Alzheimer’s, anti-epilepsy and anti-Parkinson’s) and dermocosmetic (anti-tyrosinase, anti-collagenase and anti-elastase). Table 4 summarizes the biological activities of different *Cannabis sativa* parts according to the literature.

#### 4.4.1. Antioxidant Activity

An antioxidant is a molecule that slows down or prevents the oxidation of molecules that can play an important role in an organism metabolism. *Cannabis sativa* L. proved to have plenty of antioxidant substances. The antioxidant activity of this plant has been widely reported in the literature and it was determined by using many assays, including the free radical scavenging method (DPPH), oxygen radical absorbance capacity (ORAC), ferric reducing ability of plasma (FRAP) and 2,2′-azino-bis (3-ethylbenzothiazoline-6-sulphonic acid) (ABTS), as well as other methods such as phosphomolybdenum and metal chelation. The antioxidant activity of *Cannabis sativa* has been reported in the plant seeds, leaves and aerial parts. Research on the antioxidant effects of *Cannabis sativa* L. seeds have been well reported in the literature. Manosroi, et al. [155] described the antioxidant activity of ethanolic extract using different tests, namely DPPH, chelating assay and lipid peroxidation inhibition. The obtained results showed that the seed organic extract exhibited strong antioxidant capacity, suggested by IC_50_ (low inhibitory concentration at 50%) values with 14.5 mg/mL for DPPH, 1.9 mg/mL for chelating assay and 92.7 mg/mL for lipid peroxidation inhibition assay. In another study, the methanolic extract of the seeds showed an average activity against DPPH with an inhibition value of 75% at 500 µL/mL [83]. According to these authors, this activity is probably due to the presence of polyphenols and cannabinoids, known for their strong antioxidant capacity [164,165]. Moreover, other seed metabolites, such as lignamides [156] and proteins [157], exhibited average antioxidant activities. However, they are lower than the results obtained with polyphenols and cannabinoids. Considering the high amount of polyphenols in the leaves, the extracts obtained from these plant parts have generally shown strong antioxidant activity [87,155]. For example, the hydroalcoholic extract of the leaves showed a DPPH IC_50_ value of 2.7 mg/mL [155]. This value is relatively lower than the result obtained in the seed DPPH assay, which suggests a higher antioxidant potential. Moreover, the aerial essential parts showed also excellent antioxidant activity, with values near to the positive control [163]. As previously mentioned, polyphenols are generally considered as the main group of antioxidant molecules that work through different mechanisms, such as the suppression of free radicals that initiate oxidative damage and inhibit the oxidation process, via chelation of catalytic metals or metal ions and the inhibition of lipoxygenase [166,167]. Furthermore, some volatile terpenes exhibits potent antioxidant and anti-free-radical properties [168]. This can explain the antioxidant activity of the essential oils of the aerial parts proven through different tests.

#### 4.4.2. Antimicrobial Activity

An antimicrobial is a molecule with microbicidal (kills microorganisms) or microbiostatic (slow the microbial growth and/or development) activity. These substances have different names depending on the type of targeted microorganism, such as antibacterial (for bacteria), antifungal (for fungi), antiviral (for virus) or antiparasitic (for parasites). The antimicrobial activity of essential oils and organic extracts of different parts of *Cannabis sativa* L. against several microorganisms has been reported. However, the degree of antimicrobial activity varies from cultivar to cultivar [169], as well as according to the part of the plant used, the extraction method and type of extract. The seed hydroalcoholic extract was evaluated against Gram-positive and -negative bacteria, namely *Staphylococcus aureus*, *Escherichia coli*, *Salmonella typhimurium*, *Enterobacter aerogenes*, *Enterococcus faecalis*, *Lacticaseibacillus paracasei*, *Limosilactobacillus reuteri*, *Levilactobacillus brevis*, *Lactiplantibacillus plantarum*, *Bifidobacterium bifidum*, *Bifidobacterium longum* and *Bifidobacterium breve*, and the obtained results showed low antibacterial activity with MIC values superior to 1 mg/mL [158]. Regarding antibacterial tests on Cannabis leaves, a study published by Anjum [159] compared the efficacy of four extracts obtained with acetone, chloroform, ethanol and water against three bacterial strains, namely *Escherichia coli*, *Staphylococcus aureus* and *Pseudomonas aeruginosa*. The results showed similarities between the four extracts with values near to 19 mm. Moreover, the results published by Manosroi, et al. [155] showed the effect of ethanolic extract as an antibacterial agent against *Staphylococcus mutans* with an inhibition diameter of 1.33 ± 0.58 mm. The essential oils of the aerial parts were evaluated as well for their antibacterial activity. The volatile terpenes of Cannabis exerted diverse activity intensities according to the targeted bacterial strains, where the weakest antibacterial activity was observed against *Helicobacter pylori* and *Klebsiella pneumonia* strains, with MICs values of 64 and 38 mg/mL, respectively. Moreover, a weak antibacterial activity was observed against *Micrococcus luteus* and *Staphylococcus aureus*, with an MIC of 4.7 mg/mL for both strains, whereas a more moderate inhibitory activity was observed against *Escherichia coli*, *Pseudomonas aeruginosa* and *Bacillus subtilis*, with an MIC of 1.2 mg/mL [163]. These results are explained by the fact that volatile terpenes are known to be strong antibacterial compounds, according to many biological investigations [170,171,172].

Concerning the antifungal activity, Anjum [159] compared the efficacy of four Cannabis leaf extracts, obtained with acetone, chloroform, ethanol and water, against two fungi, namely *Aspergillus niger* and *Fusarium* spp. The extracts showed similar results with inhibition diameter values ranging between 20.6 and 23 mm for *Aspergillus niger*, and 18.3 to 24.3 mm for *Fusarium* spp. In another studies, the acetone extract of Cannabis flowers showed a significant effect on the growth of *Aspergillus favus*, the flower hydroalcoholic extract led to the inhibition of 36% of fungi mycelium at concentration of 7.2 mg of dry matter per mL of culture medium [161]. Moreover, aerial part essential oils showed interesting antifungal potential. Nafis, et al. [163] reported an MIC value of 9.5 mg/mL against four fungi species, namely *Candida albicans*, *Candida glabrata*, *Candida krusei* and *Candida parapsilosis*. However, Zengin, et al. [94] discovered a weak antifungal activity against a group of clinically relevant and multidrug-resistant microorganisms belonging to *Candida* spp. and *Malassezia* spp., with essential oil MIC values superior to 12.460 µg/mL.

#### 4.4.3. Insecticidal Activity

Insecticides are active substances with the property of killing insects, their larvae and/or eggs. Insecticides act either by contact or after penetration into the digestive tract or into the respiratory system. Essential oils from inflorescences were evaluated for their mosquitocidal activities on larvae and pupae of two main malaria vectors known as *Anopheles gambiae* and *Anopheles stephensi*. The results showed that Cannabis inflorescence essential oils showed toxicity against mosquitoes with LC_50_ values of 73.5 to 78.8 ppm for *Anopheles stephensi* larvae, and 20.13 to 67.19 ppm for pupae of *Anopheles gambiae*. Their natural origin and volatile propriety makes the use of these essential oils very attractive for the formulation of stable and safe biocontrol products [162]. The cholinesterase inhibition activity had also been reported in the literature. Furthermore, a molecule purified from the ethanolic extract of Cannabis seeds, namely 3,3′-demethyl-heliotropamide, showed a moderate activity with an IC_50_ value of 46.2 µm [156], whereas the aerial part essential oils showed stronger activity against butyryl-choline esterase, with a concentration of 3.4 mg GALAE/g oil [94].

#### 4.4.4. Anticoagulant Activity

It has been suggested that plants with anticoagulant activities act as herbal remedies that could lead to the discovery of new therapeutic agents to treat thrombosis-related diseases. Blood clotting studies were conducted to determine the possible antiprothrombotic effect of Cannabis leaf metabolites, with three main cannabinoids, THC, CBD and CBN, targeted. The in vitro effect of Cannabis extract on thrombin activity was evaluated by Coetzee, et al. [160]. In their publication, two cannabinoids, namely THC and CBN, showed interesting IC_50_ values. The highest activity was obtained by THC, with a value of 1.79 mg/mL, whereas CBN showed weaker activity suggested by high IC_50_ value. However, this study also featured an in vivo test applied on obese rats in order to determine the clotting times. As a result, the Cannabis-treated rats showed an efficiency of 50% with clotting two times higher than the control groups, thus proving that cannabinoids may have a good anticoagulant activity.

#### 4.4.5. Antidiabetic Activity

Diabetes is a chronic, progressive and complex metabolic disorder characterized by abnormally high blood glucose levels. This condition is also known as hyperglycemia. Worldwide, approximately 90% of affected patients are non-insulin-dependent, classified as type 2 diabetes [173]. According to the World Health Organization (WHO), diabetes is a chronic disease that occurs when the pancreas does not produce enough insulin, or the body does not properly use the produced insulin. Insulin is a hormone that regulates the concentration of sugar in the blood. Antidiabetic activity is generally evaluated by the quantification of α-amylase inhibition [174]. This enzyme is generally produced in the pancreas and plays a key enzyme role in the increase in blood sugar by breaking down dietary carbohydrates, such as starch, into simple monosaccharides in the digestive system, followed by the further α-glucosidase degradation into glucose which, upon absorption, enters in the bloodstream. Therefore, inhibition of the enzymes α-amylase and α-glucosidase can suppress carbohydrate digestion, delay glucose absorption and, consequently, reduce blood glucose levels [175]. The study published by Zengin, et al. [94] proved that Cannabis aerial part essential oils exhibited antidiabetic properties against the α-glucosidase enzyme with a value of 3.77 mmol ACAE/g oil. This essential oil has also been evaluated against α-amylase but showed no significant result.

#### 4.4.6. Anticancer Activity

Cancer remains a major cause of morbidity and mortality worldwide. It is currently treated using classical approaches such as surgery, chemotherapy and radiotherapy. The toxic side effects associated with chemotherapy and radiotherapy often lead to adverse health effects. This explains the huge need for new drugs, safer to use with less side effects. Experimentally, several Cannabis-derived compounds demonstrated conclusive efficacy in vitro and in vivo on a wide range of cancer cell lines, including breast [176], prostate [177], cervix [178], brain [179], colon [180] and leukemia/lymphoma [181]. A number of in vitro and in vivo studies have demonstrated the effects of phytocannabinoids on tumor progression. These studies suggest that specific cannabinoids such as Δ9-THC and CBD induce apoptosis and inhibit proliferation in various cancer cell lines at concentrations ranging between 5 and 65 µm [176,182,183,184,185,186,187]. Moreover, combination of certain phytocannabinoids improved the anticancer activity of Cannabis preparations; for example, Armstrong, et al. [182] revealed that the combination of CBD and Δ9 -THC exhibited a stronger melanoma cell mortality in comparison with Δ9-THC alone. In general, phytochemicals in the Cannabis plant, and especially cannabinoids, are non-selective in their functions and limited in their differential activity on cancer cells with normal cells. Therefore, researchers show interest in isolating bioactive phytochemicals from Cannabis with potent anticancer properties and generating lead compounds based on the natural backbone of a molecule as a synthetic approach.

#### 4.4.7. Anti-Inflammatory and Analgesic Activities

Many compounds of Cannabis proved to have strong anti-inflammatory activity. The Cannabis seeds showed an inflammation-reducing capacity, especially on primary human monocytes treated with LPS. The results proved that the compounds of these seeds decreased the respective expression and secretion of IL-6 genes and TNF-α. Additionally, cannabinoids proved to be strong anti-inflammatory agents; in fact, they can suppress the production of pro-inflammatory cytokines and chemokines and may have therapeutic applications in health conditions underlying inflammatory components [188,189]. Analgesic action is defined as any procedure whose principle of activity is to reduce pain. This can be not only a drug but also any other method aimed at achieving analgesia, i.e., the abolition of the sensation of pain [190]. Clinical and experimental studies showed that Cannabis-derived compounds act as analgesic agents. However, the effectiveness of each product is variable and depends on the administration mode. With opioids being the only therapy for severe pain, the analgesic capacity of cannabinoids could provide a much-needed alternative to opioids [191]. Furthermore, cannabinoids act synergistically with opioids and act as opioid-sparing agents, allowing lower doses and fewer side effects of chronic opioid treatment [192]. Thus, the rational use of Cannabis-based medicines deserves to be seriously considered to alleviate patients’ suffering from severe pain.

#### 4.4.8. Neuroprotective Activity

The terpenes and cannabinoids of Cannabis proved also to have neuroprotective proprieties. The neuroprotective effects of 17 compounds present in the aerial parts of *Cannabis sativa* L. were evaluated in PC-12 cells including p-hydroxybenzaldehyde, (e)-methyl p-hydroxycinnamate and ferulic acid—which showed additional protective effects against H_2_O_2_-induced damage [193]. Furthermore, di Giacomo, et al. [147] reported the neuroprotective and neuromodulatory effects induced by cannabidiol and cannabigerol in rat Hypo-E22 cells and isolated hypothalamus, whereas Landucci, et al. [194] proved that appropriate concentrations of CBD or CBD/THC ratios can represent a valid therapeutic intervention in the treatment of post-ischemic neuronal death. In another study, Esposito, et al. [195] highlighted the importance of CBD as a promising new drug able to reduce neuroinflammatory responses evoked by β-amyloid. Furthermore, the study published by Perez, et al. [196] described the neuronal counting of both motor and sensory neurons after CBD treatment using immunohistochemical analysis. The obtained results showed an increase by 30% of synaptic preservation on the spinal cord for the CBD-treated group, suggesting an average neuroprotective effect.

#### 4.4.9. Antiepileptic and Anticonvulsant Activities

Despite being well-known for its psychoactive proprieties, *Cannabis sativa* L. has been investigated for additional effects on the central nervous system. A recently published study showed that CBD has a high efficacy in epilepsy with hippocampal focus than with the extrahippocampal amygdala and parvalbumin, implying a protective role in regulating hippocampal seizures and neurotoxicity at a juvenile age [197]. The efficiency of CBD has been further replicated in human populations, including adolescents and young adults with severe childhood-onset epilepsy [198]. In a 12-week open-label trial, a group of patients aged between 1 and 30 years were treated with 25 and 50 mg/kg of CBD. The treated groups showed a reduction in epileptic attack severity by 36.5%. Preclinical research has also attempted a more chronic elucidation of the efficacy of CBD as an anticonvulsant. Using the PTZ model of epilepsy, it was found that a reduction in seizure activity could be achieved with varying doses between 20 and 50 mg/kg of CBD over a 28-day treatment period [199]. Other cannabinoids, such as cannabigerol, cannabidivarin, cannabichromene, δ 9 -tetrahydrocannabinolic acid and tetrahydrocannabivarin, showed efficacy in models of Huntington’s disease and epilepsy. It is important to note that these phytocannabinoids and their combinations are warranted in a range of other neurodegenerative disorders such as Parkinson’s [200].

#### 4.4.10. Dermocosmetic Activity

Lipids in human skin play a very important role in preserving the structure of the dermis, protecting it from dehydration. However, during menopause, hormonal changes negatively affect the skin’s balance, making it more prone to developing dryness [201]. The underlying tissues, such as subcutaneous adipose tissue and muscles, undergo atrophy due to the overproduction of some enzymes—among them tyrosinase, elastase and collagenase. Tyrosinase is mainly responsible for the production of melanin on the skin, whereas collagenase and elastase target the skin structural proteins collagen and elastin and degrade them, respectively. The results published by Manosroi, et al. [155] showed that leaf extracts have anti-tyrosinase activity with an IC_50_ value of 0.07 ± 0.06 mg/mL, which suggest a strong tyrosinase inhibitory activity. Similarly, Zagórska-Dziok et al. [87] showed respective collagenase and elastase inhibitory activities of 80 and 30% at 1000 µg/mL. These activities are mainly due to the presence of polyphenols and cannabinoids.

### 4.5. Drugs Based on Cannabis sativa L.

Due to the importance of the biological evaluation’s findings, several Cannabis-based commercial pharmaceuticals have been produced. These products have many biological proprieties and were produced to treat a wide range of conditions. The first reported product has been commercialized under the name “marinol”. This drug was developed 40 years ago, in 1985, by an American pharmaceutical company, and used dronabinol and synthetic THC as active agents [202]. Likewise, “syndros” is another drug marketed in the USA in 2016 and it contains the same active ingredients [202]. Both drugs are indicated for the treatment of severe nausea and vomiting related to cancer chemotherapy and AIDS-anorexia associated with weight loss. Another active compound known as “nabilone” is used as ingredient of two drugs, “casamet” and “canemes” [203]. The “nabilone” is a synthetic analogue of THC approved by the U.S. Food and Drug Administration (U.S. FDA) for the treatment of chemotherapy and AIDS symptoms [204]. Two other cannabinoids, namely “CBD” and “THC”, are also used in two drugs commercialized under the name of Bourneville and used against two types of severe epilepsy (Lennox–Gastaut syndrome and Dravet syndrome) [205], whereas Sativex, generally known as “nabiximols”, is used to alleviate muscle spasms in multiple sclerosis disease.

## 5. Conclusions

This manuscript has reviewed and analyzed the historical, botanical, ethnopharmacological, chemical, bioinformatics and biological knowledge of *Cannabis sativa* from the earliest human communities to current medical applications, with a critical analysis of the multiple potential applications of cannabinoids in the contemporary scientific context.

At present, more than 545 phytochemicals have been described in the different parts of the Cannabis plant. The most represented metabolite class is the phytocannabinoids and they exhibit enormous structural diversity and bioactivities. *Cannabis sativa* is found in a wide variety of forms and environments on all continents and its pharmacological properties seem to go far beyond psychotic effects, with the ability to address needs such as the treatment and relief of many symptoms and diseases.

Furthermore, the relaxation of regulatory standards for therapeutic Cannabis and the conduct of more controlled clinical trials suggests that the *Cannabis sativa* plant has interesting therapeutic potential as an antiemetic, appetite stimulant in debilitating diseases (cancer and AIDS), analgesic, as well as in the treatment of multiple sclerosis, spinal cord injury, Tourette syndrome, epilepsy and glaucoma. Further clinical research is needed to investigate the potential therapeutic uses of this plant in specific medical conditions. Scientifically designed trials will help establish which of the cannabinoids produce the various beneficial effects described, or whether these are the result of a combination of cannabinoids. The research would also help to better characterize the adverse effects of each cannabinoid.

## Figures and Tables

**Figure 1 plants-12-01245-f001:**
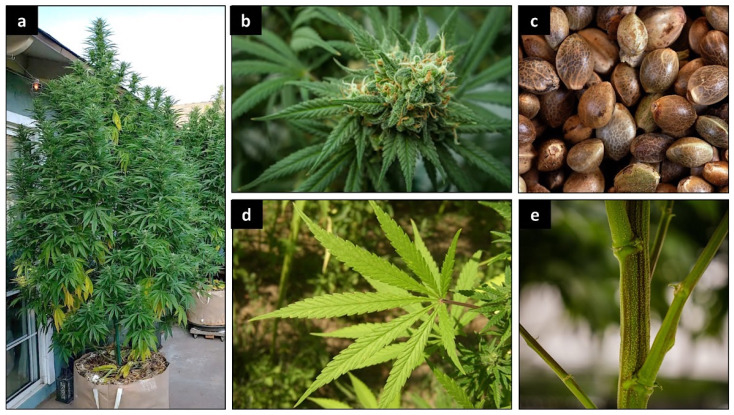
*Cannabis sativa* L. General aspect (**a**); inflorescence (**b**); seed (**c**); leaf (**d**); stem (**e**).

**Figure 2 plants-12-01245-f002:**
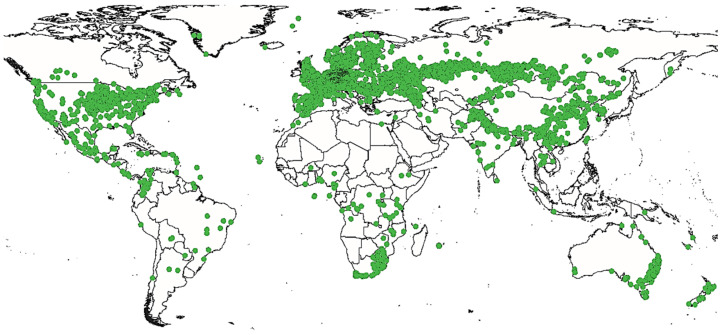
Geographic distribution of *Cannabis sativa* L.

**Table 1 plants-12-01245-t001:** Traditional uses of different parts of *Cannabis sativa*.

Plant Part	Traditional Use	Preparation	Administration	Reference
Seed	Nutrition	Powder	Oral	[59]
Seed	Narcotic, painkiller, treat nausea and vomiting, stimulate appetite in AIDS patients, hepatitis C, anxiety, seizure, muscle relaxants, anticancer and weight control	ND	ND	[60]
Seed	Hair fortification	Powder	External	[61]
Seed	Analgesic, antiarthritic and antirheumatic	Oil	External	[62]
Leaf	Eczema	Powder	External	[61]
Leaf	Bloating, cough, mucus	Leaf juice	Oral	[63]
Leaf	Central nervous system (CNS) depressant, gout, arthritic pain	Powder	Oral	[64]
Leaf	Schizophrenia-like psychotic problems	Oil	External	[65]
Leaf	Gastric disorders	ND	ND	[66]
Leaf	Skin and subcutaneous tissue disorders, circulatory system and blood disorders	Juice, paste or powder	ND	[62]
Inflorescence	Sedative, dysentery, diarrhea and appetite loss	ND	Oral	[62]
Stem	Firewood or torch wood	Raw	ND	[62]
Stem	Construction, materials, dress, papermaking, making ropes	Raw	ND	[5]
Seed, flower	Hair care	ND	ND	[67]
Leaf, inflorescence	Soporific, abortifacient	ND	ND	[68]
Leaf, root	Cancer, hypertension, antidote to poison, itch, rheumatoid arthritis	ND	ND	[66]
Root	Fever	Maceration	External	[69]
Root	Gout, arthritis	Boiled roots	External (cataplasm)	[70]
Root	Joint pain	Decoction	External (cataplasm)	[70]
Root	Skin burns	Raw or decoction mixed with butter	External (topical)	[70]
Root	Inflammation	Boiled roots, decoction	External (cataplasm)	[70]
Root	Childbirth, postpartum, hemorrhage	Decoction	Oral	[70]
Root	Erysipelas, toxins and infections	Pulverized and mixed with wine	Oral/external	[70]
Whole plant	Pain, gastric disorders, diabetes, scars and asthma	ND	ND	[71]

**Table 2 plants-12-01245-t002:** Chemical composition of *Cannabis sativa* different plant parts.

Plant Part	Extraction Method (Type of Extract)	Class of Metabolites	Compounds (Quantities)	Reference
Seed	Maceration (hexane)	Fatty acids	Linoleic acid (47.06%) Oleic acid (43.20%) Palmitic acid (4.88%) Stearic acid (3.32%)	[80]
Seed	Soxhlet (petroleum benzine)	Fatty acids	Linoleic acid (58.41 ± 0.04%) α-Linolenic Acid (16.26 ± 0.03%) Oleic acid (16.05 ± 0.02%) Palmitic acid (5.59 ± 0.12%) Stearic acid (2.46 ± 0.01%)	[81]
Seed	Soxhlet (ethanol–water 80:20)	Polyphenols	Gallic acid (12.9 ± 18.28 mg/100 g) (+)-Catechin (5.995 ± 5.23 mg/100 g) 1,2-Dihydroxybenzene (5.155 ± 4.59 mg/100 g) 3,4-Dihydroxybenzoic acid (4.89 ± 4.68 mg/100 g) Syringic acid (3.795 ± 1.99 mg/100 g) Caffeic acid (2.475 ± 3.53 mg/100 g) Quercetin (2.08 ± 3.36 mg/100 g) Rutin trihydrate (0.915 ± 1.15 mg/100 g) Isorhamnetin (0.765 ± 0.89 mg/100 g) trans-Ferulic acid (0.685 ± 0.61 mg/100 g) Apigenin-7-glucoside (0.55 ± 0.38 mg/100 g) Naringenin (0.255 ± 0.33 mg/100 g) trans-Cinnamic acid (0.19 ± 0.24 mg/100 g) Resveratrol (0.17 ± 0.24 mg/100 g) p-Coumaric acid (0.165 ± 0.17 mg/100 g)	[81]
Seed	Ultrasound-assisted extraction (ethanol)	Polyphenols	Cannabisin A (105.1 ± 54 mg/100 g DW) N-trans-caffeoyltyramine (49 ± 34.2 mg/100 g DW) Cinnamic acid (3.75 ± 3.55 mg/100 g DW) *p*-hydroxybenzoic acid (2.1 ± 0.9 mg/100 g DW) Protocatechuic acid (1 ± 0.6 mg/100 g DW)	[82]
Seed	Maceration (methanol–water 80:20)	Polyphenols	Quercetin-O-glucoside (ND) N-trans-caffeoyltyramine (ND) Rutin (ND)	[83]
Seed	Maceration (methanol–water 80:20)	Polyphenols	Cannabisin A, B, C, D, E, F, G, I, M, N, O (ND)	[83]
Seed	Soxhlet (hexane)	Tocopherols	γ-tocopherol (426 mg/kg) δ-tocopherol (33 mg/kg) α-tocopherol (13 mg/kg) β-tocopherol (2 mg/kg)	[84]
Seed	Ultrasound-assisted extraction (ethanol)	Tocopherols	γ-tocopherol (7.95 ± 3.35 mg/100 g DW) δ-tocopherol (0.95 ± 0.35 mg/100 g DW)	[82]
Seed	Maceration (hexane)	Phytosterols	β-Sitosterol (90.75 ± 0.42%) Campesterol (6.20 ± 0.00%) Stigmasterol (2.88 ± 0.17%)	[85]
Seed	Soxhlet (hexane)	Phytosterols	β-sitosterol (68.0%) Campesterol (17.1%) Δ-5-avenasterol (7.8%) Stigmasterol (3.8%) Δ-5,25-stigmastadienol (1.1%)	[84]
Seed	Ultrasound-assisted extraction (ethanol)	Carotenoids	Lutein (2.45 ± 0.95 mg/100 g DW) β-Carotene (0.5 ± 0.3 mg/100 g DW)	[82]
Seed	Supercritical fluid extraction (CO_2_)	Aldehydes	Hexanal (39.57 ± 0.91 mg/kg) Octadienal (10.29 ± 3.18 mg/kg) Heptadienal (9.38 ± 1.41 mg/kg) Nonenal (8.77 ± 1.27 mg/kg) Nonanal (8.34 ± 1.28 mg/kg)	[85]
Seed	Supercritical fluid extraction (CO_2_)	Alcohols	Hexanol (30.66 ± 0.95 mg/kg)	[85]
Seed	Maceration (hexane)	Hydrocarbons	Dodecane (112.5 ± 2.06 mg/kg) Tetradecane (69.0 ± 1.40 mg/kg) 1.3-Di-tert-butylbenzene (46.6 ± 2.25 mg/kg)	[85]
Leaf	Hydrodistillation	Terpenes	(E)-Caryophyllene (28.3 ± 4.1%) α-Humulene (9.3 ± 1.1%) β-Selinene (4.7 ± 0.9%) Caryophyllene oxide (4.3 ± 0.9%) α-Selinene (3.1± 0.6%)	[86]
Leaf	Ultrasound-assisted extraction (methanol)	Polyphenols	Apigenin C-(hexoside-O-rhamnoside) (0.83 mg/g) Luteolin C-(hexoside-O-rhamnoside) (0.67 mg/g) Luteolin di-C-hexoside (0.60 mg/g) Luteolin glucuronide(0.60 mg/g) Apigenin di-C-hexoside (0.54 mg/g)	[86]
Leaf	Ultrasound-assisted extraction (methanol)	Cannabinoids	CBD (11.2 ± 1.9%) CBDV (0.8 ± 0.2%) THC (0.7 ± 0.2%) CBC (0.5 ± 0.1%)	[86]
Leaf	Ultrasound-assisted extraction (water–ethanol 20:80)	Cannabinoids	Cannabidiol acid (150.00 ± 16.84 mg/g DW) Cannabidiol (31.00 ± 2.86 mg/g DW) THCA (6.50 ± 0.52mg/g DW) Cannabigerolic acid (6.30 ± 0.52 mg/g DW) Δ^9^-THC (4.00 ± 0.34 mg/g DW)	[87]
Leaf	Maceration (ethanol–acetic acid 90:10)	Alkaloids	Cannabisativine (410.30 μg/g) Cannabimine C (376.12 μg/g) Anhydrocannabisativine (218.11 μg/g) Aconitine (160.43 μg/g) Boldine (103.41 μg/g) Strychnine (72.63 μg/g)	[88]
Leaf	Vacuum liquid chromatography (ethanol–water 95:05)	Stilbenoids	Canniprene (ND); Combretastatin B-2 (ND) α,α′-dihydro-3,4′,5-trihydroxy-4,5′-diisopentenylstilbene (ND)	[89]
Flower	Hydrodistillation	Terpenes	(E)-Caryophyllene (28.5 ± 3.1%) α-Humulene (9.2 ± 1.7%); β-Selinene (4.3 ± 0.8%); α-Selinene (2.9 ± 0.6%) Caryophyllene oxide (2.3 ± 0.5%); α-trans-Bergamotene (1.9 ± 0.4%)	[86]
Flower	Ultrasound-assisted extraction (methanol)	Polyphenols	Quercetin di-C-hexoside (2.55 mg/g) Luteolin C-hexoside-2″-O-hexoside (1.01 mg/g) Apigenin di-C-hexoside (0.68 mg/g) Apigenin C-(hexoside-O-rhamnoside) (0.51 mg/g)	[86]
Flower	Ultrasound-assisted extraction (methanol)	Cannabinoids	CBD (24.9 ± 3.9%) THC (1.4 ± 0.3%) CBDV (1.4 ± 0.3%) CBTC (0.9 ± 0.2%)	[86]
Inflorescence	Hydrodistillation	Terpenes	Transcaryophyllene (38.2 ± 1.7%) Nerolidol (12.7 ± 1.2%) α-pinene (11.8 ± 0.4%) β-pinene (3.4 ± 0.3%) Cedrol (2.2 ± 0.0%) Myrcene (1.7 ± 0.5%) α-bisabolol (0.6 ± 0.1%) γ-terpinene (0.5 ± 0.3%) Camphene (0.2 ± 0.0%) α-humulene (0.2 ± 0.0%) α-terpinene (0.1 ± 0.0%) Menthol (0.1 ± 0.0%)	[90]
Inflorescence	Hydrodistillation	Terpenes	β-caryophyllene (14.4 ± 0.89%) Caryophyllene oxide (7.0 ± 1.06%) α-humulene (5.3 ± 0.10%) Selina-3,7(11)-diene (3.4 ± 0.33%) α-pinene (3.0 ± 0.04%) Myrcene (2.6 ± 0.16%) 14-hydroxy-9-epi-(E) -caryophyllene (2.5 ± 0.12%) Humulene oxide II (2.4 ± 0.24%) Caryophylla-4(14),8(5)-dien-5-ol (1.4 ± 0.15%) β-selinene (1.2 ± 0,07%) α-bisabolol (1.1 ± 0.13%) Selin-6-en-4-ol (0.8 ± 0.23%)	[91]
Inflorescence	Maceration (methanol)	Cannabinoids	CBDA (9.515 ± 1.085%) tCBD (8.695 ± 0.955%) tTHC (0.545 ± 0.075%) CBGA (0.535 ± 0.365%) CBG (0.535 ± 0.355%) tCBG (0.490 ± 0.080%) THCA (0.490 ± 0.080%) CBD (0.345 ± 0.005%) Δ^9^-THC (0.080 ± 0.000%) Δ^8^-THC (0.045 ± 0.005%)	[90]
Inflorescence	Microwave-assisted hydrodistillation	Cannabinoids	THCA (0.66 ± 0.04%) THC (0.34 ± 0.02%) CBDA (0.05 ± 0.005%)	[92]
Leaf and flower	Soxhlet (methanol–water 75:25)	Flavonoids	Luteolin-O-β-D-glucuronide (3.75 ± 0.75 mg/g DW) Apigenin-O-β-D-glucuronide (1.25 ± 0.25 mg/g DW) Vitexin (1.25 ± 0.25 mg/g DW)	[93]
Leaf and inflorescence	Hydrodistillation	Terpenes	(E)-Caryophyllene (28%) Caryophyllene oxide (15%) Humulene (13%) β-Myrcene (11%) α-Pinene (8%)	[94]
Leaf and inflorescence	Hydrodistillation	Phenolic compounds	Naringenin (706 µg/mL) Naringin (83 µg/mL) Catechin (60 µg/mL) Epicatechin (56 µg/mL)	[94]
Leaf and stem	Rapid solid–liquid dynamic extraction (ethanol)	Terpenes	Caryophyllene (52.78 ± 2.61%) Humulene (13.49 ± 0.14%) Linalool (9.42 ± 0.24%) α-bergamotene (6.14 ± 0.51%) cis-β-farnesene (3.54 ± 0.42%) Aromadendrene (2.86 ± 0.12%)	[95]
Leaf and stem	Rapid solid–liquid dynamic extraction (ethanol)	Polyphenols	Luteolin (304.37 ± 1.10 µg/g DW) Ferulic acid (247.77 ± 0.64 µg/g DW) Gallic acid (52.29 ± 0.98 µg/g DW) Apigenin (51.43 ± 0.48 µg/g DW) p-OH-benzoic acid (47.70 ± 0.75 µg/g DW) Rosmarinic acid (27.09 ± 0.85 µg/g DW)	[95]
Resin	Supercritical fluid (CO_2_)	Cannabinoids	CBD (72.12 μg/mL) THC (48.02 μg/mL) CBC (4.78 μg/mL) CBDA (2.34 μg/mL) CBN (0.40 μg/mL)	[96]
Resin	Ultrasound-assisted extraction (ethanol)	Cannabinoids	Δ^9^-THC > 20%	[97]

Cannabidiol (CBD); cannabidivarine (CBDV); cannabicitran (CBTC); cannabigerol (CBG); cannabichromene (CBC); cannabinol (CBN); δ9-tetrahydrocannabinol (δ^9^-THC); cannabidiolic acid (CBDA); tetrahydrocannabinolic acid (THCA); delta-8-tetrahydrocannabinol (δ^8^-THC); cannabigerol (CBG); cannabigerol acid (CBGA); cannabielsoin (CBE); cannabicitran (CBTC); cannabiripsol (CBR); total cannabidiol (tCBD); total cannabigerol (tCBG); total tetrahydrocannabinol (tTHC).

**Table 3 plants-12-01245-t003:** Molecular docking of *Cannabis sativa* L. compounds.

Enzyme				Ligand		Docking Tool	Score (kcal/mol)	Reference
Name	ID (pdb = *p* / Unipr = *up*)	Class of Protein	Biological Activity	Name	Class of Metabolite			
Acetylcholine esterase (ACHE)	1EVE *(p)*	Hydrolase	Pesticidal	Cannabioxepane	Cannabinoids	AutoDock vina	–10.4	[127]
				Δ-9-THCA	Cannabinoids		–10.3	
				Δ-8-THC	Cannabinoids		–10.1	
				CBN	Cannabinoids		–10.1	
				CBT	Cannabinoids		–9.8	
				CBD	Cannabinoids		–9.8	
				CBL	Cannabinoids		–9.6	
				Isocannabispiradienone	Cannabinoids		–9.4	
				CBC	Cannabinoids		–9.4	
				CBCA	Cannabinoids		–9.3	
				CBGA	Cannabinoids		–9.2	
				Cannabispiradienone	Terpenes		–9.2	
				Cannabispirol	Polyphenols		–9.2	
Butyrylcholinesterase (BCHE)	1P0I *(p)*	Hydrolase	Pesticidal	Cannabioxepane	Cannabinoids	AutoDock vina	–9.8	[127]
				CBL	Cannabinoids		–8.9	
				CBN	Cannabinoids		–8.8	
				CBT	Cannabinoids		–8.7	
				Δ-8-THC	Cannabinoids		–8.7	
				CBL	Cannabinoids		–8.4	
				Isocannabispiradienone	Cannabinoids		–8.3	
				CBD	Cannabinoids		–8.2	
				Cannabispiradienone	Terpenes		–8.2	
				Cannabispiran	Polyphenols		–8.2	
Acetylcholine esterase (ACHE)	1EVE *(p)*	Hydrolase	Pesticidal	CBD	Cannabinoids	AutoDock vina	−14.38	[128]
				CBN	Cannabinoids		−13.91	
				Δ-9-THC	Cannabinoids		−13.82	
*Plasmodium falciparum* dihydrofolate reductase-thymidinesynthase (pfdhfr-ts)	1J3I *(p)*	Lyase	Antimalarial	7-oxo-9a-hydroxyhexahydrocannabinol	Cannabinoids	AutoDock vina	−9.40	[129]
				10ar-hydroxyhexahydrocannabinol	Cannabinoids		−9.20	
				10-oxo-delta6a,10a-tetrahydrocannabinol	Cannabinoids		−9.20	
				8-oxo-delta9-tetrahydrocannabinol	Cannabinoids		−9.10	
				10a-hydroxyhexahydrocannabinol	Cannabinoids		−9.10	
*Leishmania major* pteridine reductase 1 (lmajptr1)	1E7W, 1W0C, 2BF7 and 3H4V *(p)*	Oxido-reductase	Anti-leishmania	4,5,4’5’-dimethylenedioxy-3,3’-dimethoxy-7,7’-epoxylignan	Polyphenols	Molegro virtual docker	−35.01	[130]
				Cannflavin A	Flavonoids		−34.42	
*Leishmania mexicana* glycerol-3-phosphate dehydrogenase (lmexgpdh)	1EVZ, 1M66, 1N1E and 1N1G *(p)*	Oxido-reductase	Anti-leishmania	4-terpenylcannabinolate	Cannabinoids	Molegro virtual docker	−35.90	[130]
				3’-o-methyldiplacone	Flavonoids		−35.44	
				4’-o-methyldiplacone	Flavonoids		−34.18	
				Sophoronol E	Flavonoids		−33.89	
*Leishmania major* methionyl-transynthetase (lmajmetrs)	3KFL *(p)*	Ligase	Anti-leishmania	4,6-dibenzoyl-2-[phenylhydroxymethyl]-3(2h)-benzofuranone	Polyphenols	Molegro virtual docker	−38.81	[130]
*Leishmania donovani* cyclophilin (ldoncyp)	2HAQ and 3EOV *(p)*	Isomerase	Anti-leishmania	3’-o-methyldiplacone	Flavonoids	Molegro virtual docker	−30.23	[130]
*Leishmania major* oligopeptidase b (lmajopb)	2XE4 *(p)*	Protease	Anti-leishmania	3-methoxycitrunobin-4-methyl ether	Polyphenols	Molegro virtual docker	−30.74	[130]
				4’-o-methylglycyrrhisoflavone	Flavonoids		−30.47	
*Leishmania major* uridine 2p-glucose pyrophosphorylase (lmajugpase)	2OEF and 2OEG *(p)*	Transferase	Anti-leishmania	4’,6-dihydroxy-2-[phenylmethylene]-3(2h)-benzofuranone	Polyphenols	Molegro virtual docker	−33.94	[130]
*Leishmania major* n-myristoyl-transferase (lmajnmt)	2WSA, 3H5Z and 4A30 *(p)*	Transferase	Anti-leishmania	Diplacone	Flavonoids	Molegro virtual docker	−32.43	[130]
				Trans-4-isopentenyl-3,5,2,4 -tetrahydroxystilbene	Polyphenols		−30.45	
*Leishmania infantum* nicotinamidase (linfpnc1)	3R2J *(p)*	Hydrolase	Anti-leishmania	5,8-dihydroxy-1-hydroxymethylnaphtho-[2,3-c] furan-4,9-dione	Polyphenols	Molegro virtual docker	−23.45	[130]
				Umckalin	Polyphenols		−21.85	
				Scoparone	Flavonoids		−21.77	
*Leishmania major* dihydroorotate dehydrogenase (lmajdhodh)	3GYE, 3MHU and 3MJY *(p)*	Oxido-reductase	Anti-leishmania	Aristolignin	Flavonoids	Molegro virtual docker	−31.21	[130]
				Crotaorixin	Flavonoids		−31.84	
				Mammea B/BA	Polyphenols		−29.09	
*Leishmania mexicana* pyruvate kinase (lmexpyk)	1PKL, 3HQP and 3PP7 *(p)*	Protein kinase	Anti-leishmania	Kusunokinin	Polyphenols	Molegro virtual docker	−31.19	[130]
*Leishmania major* phosphodiesterase 1 (lmajpde1)	2R8Q *(p)*	Hydrolase	Anti-leishmania	Machaeriol B	Cannabinoids	Molegro virtual docker	−28.97	[130]
*Leishmania major* tyrosyl-trna synthetase (lmajtyrrs)	3P0H and 3P0J *(p)*	Ligase	Anti-leishmania	Bractein triacetate	Polyphenols	Molegro virtual docker	−33.08	[130]
COVID-19 protease	6LU7 *(p)*	Protease	Antiviral	CBD	Cannabinoids	AutoDock vina	−7.10	[131]
Hiv-1 protease	*ND*	Protease	Antiviral	Cannflavin	Flavonoids	AutoDock vina	−9.70	[132]
Human angiotensin-converting enzyme (ACE2)	1R4L *(p)*	Protease	Antiviral	THC	Cannabinoids	AutoDock vina	−9.2	[133]
2019-ncov spike protein s2 subunit	6LXT *(p)*	Surface glycoprotein	Antiviral	THC	Cannabinoids	AutoDock vina	−4.2	[133]
SARS-CoV-2 mpro	ND	Protease	Antiviral	Cannabisin A	Lignanamide	AutoDock vina	−12.76	[134]
3c-like protease (C)	6LU7 *(p)*	Protease	Antiviral	Hesperidin	Flavonoids	AutoDock vina	−8.3	[135]
				Nabiximols	Cannabinoids		−8	
Spike glycoprotein (S)	6VXX *(p)*	Surface glycoprotein	Antiviral	Hesperidin	Flavonoids	AutoDock vina	−10.4	[135]
				Nabiximols	Cannabinoids		−10.2	
Inhibitor of nuclear factor kappa-b kinase subunit β (IKKbeta)	3BRT *(p)*	Inhibitor kappa kinase	Anti-inflammatory	CBD	Cannabinoids	AutoDock-tools	−7.99	[136]
Mitogen-activated protein kinase 14	4IDT *(p)*	Map kinase	Anti-inflammatory	CBD	Cannabinoids	AutoDock-tools	−7.35	[136]
Cellular tumor antigen p53	IAIE *(p)*	Tumor suppressor	Anti-inflammatory	CBD	Cannabinoids	AutoDock-tools	−6.08	[136]
Nf-kappa-b inhibitor alpha	1IKN *(p)*	Nfkb inhibitor	Anti-inflammatory	CBD	Cannabinoids	AutoDock-tools	−5.82	[136]
Tnf receptor-associated factor 6	1IB6 *(p)*	Traf protein	Anti-inflammatory	CBD	Cannabinoids	AutoDock-tools	−5.74	[136]
Transcription factor p65	1NFI *(p)*	Nfkb inhibitor	Anti-inflammatory	CBD	Cannabinoids	AutoDock-tools	−5.66	[136]
Epidermal growth factor receptor	1MOX *(p)*	Epidermal growth factor	Anti-inflammatory	CBD	Cannabinoids	AutoDock-tools	−5.64	[136]
Nf-kappa-b essential modulator	3BRV *(p)*	Nfkb inhibitor	Anti-inflammatory	CBD	Cannabinoids	AutoDock-tools	−5.36	[136]
Rac-alpha serine/threonine-protein kinase	1UNQ *(p)*	Transferase	Anti-inflammatory	CBD	Cannabinoids	AutoDock-tools	−5.34	[136]
Mitogen-activated protein kinase 3	2O2V *(p)*	Map kinase	Anti-inflammatory	CBD	Cannabinoids	AutoDock-tools	−5.34	[136]
Poly [adp-ribose] polymerase 1	2COK *(p)*	DNA polymerase	Anti-inflammatory	CBD	Cannabinoids	AutoDock-tools	−4.88	[136]
Hypoxia-inducible factor 1-alpha	1H2K *(p)*	Hypoxia inducible factor	Anti-inflammatory	CBD	Cannabinoids	AutoDock-tools	−4.41	[136]
Inhibitor of nuclear factor kappa-b kinase subunit a	3BRT *(p)*	Inhibitor kappa kinase	Anti-inflammatory	CBD	Cannabinoids	AutoDock-tools	−4.32	[136]
Nuclear factor kappa-b p105 subunit	IMDI *(p)*	Nfkb inhibitor	Anti-inflammatory	CBD	Cannabinoids	AutoDock-tools	−4.23	[136]
G1/s-specific cyclin-d1	5VZU *(p)*	Proto-oncogene regulator	Anti-inflammatory	CBD	Cannabinoids	AutoDock-tools	−3.44	[136]
Activator protein-1 (AP-1)	1JUN *(p)*	Transcription factor	Anti-inflammatory	CBD	Cannabinoids	AutoDock-tools	-3.41	[136]
Crystal structure of the DLC1 RhoGAP domain in liver cancer 1	3KUQ *(p)*	GTPase-activating proteins	Anticancer	CBC	Cannabinoids	Docking-server	−3.89	[137]
				THCV	Cannabinoids		−3.25	
				CBGV	Cannabinoids		−3.21	
				CBDA	Cannabinoids		−3.34	
Placental aromatase cytochrome p450	3EQM *(p)*	Cytochrome	Anticancer	CBDC1	Cannabinoids	Molegro (Glide)	−9.03	[138]
				CBGV	Cannabinoids		−7.8	
				CBCA	Cannabinoids		−7.73	
				CBCVA	Cannabinoids		−7.45	
				CBCV	Cannabinoids		−8.29	
				CBDV	Cannabinoids		−8.34	
				CBT	Cannabinoids		−7.86	
				(Δ−9-THC)	Cannabinoids		−7.43	
				CBR	Cannabinoids		−6.93	
Heat shock protein 90 (hsp90)	2QG2 *(p)*	Chaperone protein	Anticancer	Guaiol	Terpenes	AutoDock vina	−10.80	[139]
Topoisomerase II alpha	5GWK *(p)*	Nuclear enzyme	Anticancer	7-o-methylcyanidin	Flavonoids	Molegro (Glide)	−10.395	[140]
				Rutin	Flavonoids		−9.847	
				Luteolin-7-o-glucoside	Flavonoids		−9.563	
				Myricetin 7-glucoside	Flavonoids		−9.383	
Tumor necrosis factor-α (TNF-α)	2E7A *(p)*	Tumor necrosis factor	Anticancer	Ascorbic acid	Vitamins	AutoDock vina	−5.4	[141]
				Linoleic acid	Terpenes		−3.8	
				Tryptophan	Amino acids		−5.6	
Arachidonate 5-lypoxygenase	5IR4 (*p*)	Lipoxygenase	Anticancer and anti-inflammatory	Δ-9-THC	Cannabinoids	Molecular operating environment	−4.57	[142]
				Δ-8-THC	Cannabinoids		−4.87	
				CBC	Cannabinoids		−5.14	
				CBG	Cannabinoids		−5.59	
				CBL	Cannabinoids		−4.83	
				CBD	Cannabinoids		−4.97	
Human myeloperoxidase (MPO)	3F9P *(p)*	Peroxidase	Anti-inflammatory and degenerative processes	Peptide YGRDEISV	Proteins	Discovery studio	−114.6	[143]
				Peptide LDLVKPQ	Proteins		−82.8	
Angiotensin-converting enzyme (ACE)	1O8A *(p)*	Protease	Hypertension	Peptide WVYY	Proteins	Accelrys discovery studio	−27.25	[144]
				Peptide WYT	Proteins		−21.99	
Renin	2V0Z *(p)*	Protease	Hypertension	Peptide SVYT	Proteins	Accelrys discovery studio	−25.33	[144]
				Peptide WYT	Proteins		−19.12	
Angiotensin-converting enzyme (ACE)	1O8A *(p)*	Protease	Hypertension	Peptide ALVY	Proteins	Accelrys discovery studio	−69.23	[145]
				Peptide LLVY	Proteins		−65.97	
				Peptide LSTSTDVR	Proteins		−105.59	
				Peptide LLAPHY	Proteins		−86.86	
Cannabinoid receptor 1 (CNR1)	P21554 *(up)*	G-protein coupled receptor	Epilepsy	8b-hydroxy-δ9-trans-tetrahydrocannabinolate	Cannabinoids	Molegro (Glide)	−8.039	[146]
			Epilepsy	10aa-hydroxy-10-oxo-d8-tetrahydrocannabinol	Cannabinoids		−8.52	
			Epilepsy	10aα-hydroxyhexahydrocannabinol	Cannabinoids		−8.55	
			Epilepsy	CBNM	Cannabinoids		−8.761	
			Epilepsy	Cannabichromanone D	Cannabinoids		−8.446	
			Epilepsy	CBL	Cannabinoids		−8.336	
			Epilepsy	5-acetoxy-6-geranyl-3-npentyl-1,4-benzoquinone	Cannabinoids		−8.571	
			Epilepsy	CBTC	Cannabinoids		−8.574	
Androgen receptor (AR)	P10275 *(up)*	Nuclear receptor	Epilepsy	CBNM	Cannabinoids	Molegro (Glide)	−8.706	[146]
			Epilepsy	5-acetoxy-6-geranyl-3-npentyl-1,4-benzoquinone	Cannabinoids		−8.262	
			Epilepsy	CBTC	Cannabinoids		−8.015	
Glycogen synthase kinase-3 beta (GSK3B)	P49841 *(up)*	Protein kinase	Epilepsy	8a-hydroxy-δ9-trans-tetrahydrocannabinolate	Cannabinoids	Molegro (Glide)	−6.439	[146]
			Epilepsy	8b-hydroxy-δ9-trans-tetrahydrocannabinolate	Cannabinoids		−5.876	
			Epilepsy	CBL	Cannabinoids		−6.183	
Albumin	P02768 *(up)*	Albumin	Epilepsy	(-)-trans-10-ethoxy-9-hydroxy-d6a(10a)-tetrahydrocannabinol	Cannabinoids	Molegro (Glide)	−6.429	[146]
			Epilepsy	CBNM	Cannabinoids		−6.306	
			Epilepsy	CBL	Cannabinoids		−6.057	
Neurokinin 3 receptor (NK3R)	1F88 *(p)*	Neurokinin receptor	Neuro-protective	CBD	Cannabinoids	AutoDock	−6.72	[147]
				CBG	Cannabinoids		−10.36	
Angiotensin-converting enzyme 2 (ACE2)	6CS2 *(p)*	Protease	Neuro-protective	CBD	Cannabinoids	AutoDock vina	−8.9	[148]
Interleukin-6 (IL6)	1ALU *(p)*	Cytokine	Neuro-protective	CBD	Cannabinoids	AutoDock vina	−8.2	[148]
Transmembrane protease serine 2 (TMPRSS2)	3NPS *(p)*	Protease	Neuro-protective	CBN	Cannabinoids	AutoDock vina	−8.7	[148]
Nrp1 protein	7BP6 *(p)*	Semaphorin receptor	Neuro-protective	CBN	Cannabinoids	AutoDock vina	−8.5	[148]
Tyrosine phosphatase-1b (PTP1B)	1NWE *(p)*	Hydrolase	Dermo-cosmetic	Chrysophanol	Polyphenols	Charmm-based docker	−24.34	[149]
P glycoprotein (P-GP)	4Q9H *(p)*	GTPase	Multi-drug resistance	Cannabisin M	Lignanamide	AutoDock vina	−10.2	[150]
				Cannabisin N	Lignanamide		−10.2	
				Cannabisin A	Lignanamide		−10.1	
				Cannabisin B	Lignanamide		−10.1	
				Cannabisin C	Lignanamide		−10.1	
				Cannabisin D	Lignanamide		−10.1	

Cannabichromene (CBC); cannabichromenic acid (CBCA); cannabichromevarin (CBCV); cannabichromevarinic acid (CBCVA); cannabicitran (CBTC); cannabicoumaronone (CBCN); cannabicyclol (CBL); cannabidiol (CBD); cannabidiol-c4 (CBDC4); cannabidiolic acid (CBDA); cannabidiorcol (CBDC); cannabidivarin (CBDV); cannabielsoin (CBE); cannabielsoin (CBL); cannabigerol (CBG); cannabigerolic acid (CBGA); cannabigerovarin (CBGV); cannabinodiol (CBND); cannabinodivarin (CBVD); cannabinol (CBN); cannabinol methyl ether (CBNM); cannabiripsol (CBR); cannabitriol (CBT); cannabivarin (CVN); tetrahydrocannabinol (THC); tetrahydrocannabivarin (THCV); δ-8-tetrahydrocannabinol (Δ-8-THC); δ-9-tetrahydrocannabinol (Δ-9-THC); δ9-tetrahydrocannabinolic acid (Δ-9-THCA).

**Table 4 plants-12-01245-t004:** Biological activities of different parts of *Cannabis sativa* L.

Plant Part	Extraction Method (Solvent)	Bioactive Metabolites	Biological Activity	Results	Reference
Seed	Ultrasound-assisted extraction (methanol–water 80:20)	**Polyphenols**	**Antioxidant**—DPPH **Standard:** (not used)	Inhibition activity = 74 ± 1% at 500 µL/mL	[83]
Seed	Maceration (ethanol–water 95:05)	**Cannabinoids and polyphenols**	**Antioxidant**—DPPH **Standard:** Ascorbic acid (IC_50_ = 0.012 ± 0.002 mg/mL)	IC_50_ = 14.39 ± 2.27 mg/mL	[155]
Seed	Maceration (ethanol–water 95:05)	**Cannabinoids and polyphenols**	**Antioxidant**—metal ion chelating assay **Standard:** EDTA (CC_50_ = 0.15 ± 0.002 mg/mL)	CC_50_ = 1.92 ± 1.05 mg/mL	[155]
Seed	Maceration (ethanol–water 95:05)	**Cannabinoids and polyphenols**	**Antioxidant**—lipid peroxidation inhibition **Standard:** α-tocopherol (IPC_50_ = 0.045 ± 0.002 mg/mL)	IPC_50_ = 92.68 ± 30.77 mg/mL	[155]
Seed	Maceration (ethanol–water 75:25)	**Lignanamides**	**Antioxidant**—DPPH **Standard:** Quercetin (IC_50_ = 25.5 µm)	**Cannabisin M** (IC_50_ = 69.5 µm) **Cannabisin N and O** (IC_50_ = ND) **3,3′-demethyl-heliotropamide** (IC_50_ = 39.3 µm)	[156]
Seed	Maceration (ethanol–water 75:25)	**Lignanamides**	**Antioxidant**—ORAC **Standard:** Quercetin (IC_50_ = 0.40 µm)	**Cannabisin M** (IC_50_ = 6.61 µm) **Cannabisin N and O** (IC_50_ = ND) **3,3′-demethyl-heliotropamide** (IC_50_ = 0.56 µm)	[156]
Seed	Maceration (ethanol–water 75:25)	**Lignanamides**	**Antioxidant**—ABTS **Standard:** Quercetin (IC_50_ = 9.19 µm)	**Cannabisin M** (IC_50_ = 74.70 µm) **3,3′-demethyl-heliotropamide** (IC_50_ = 16.41 µm)	[156]
Seed oil	Acid/alkali extraction	**Proteins**	**Antioxidant**—DPPH **Standard:** (not used)	**Alkali soluble proteins:** - inhibition value = 73.33% after ht enzyme hydrolysis. **Acid soluble proteins:** - inhibition value = 68.67% after ht enzyme hydrolysis.	[157]
Seed flour	Ultrasound-assisted extraction (methanol–water 80:20)	**Polyphenols**	**Antioxidant**—DPPH **Standard:** (not used)	Inhibition activity = 67 ± 1% at 500 µL/mL	[83]
Seed oil	Ultrasound-assisted extraction (methanol–water 80:20)	**Polyphenols**	**Antioxidant**—DPPH **Standard:** (not used)	Inhibition activity = 22 ± 2% at 500 µL/mL	[112]
Seed	Maceration (ethanol–water 80:20)	**Polyphenols**	**Antibacterial** (against *Staphylococcus aureus, Escherichia coli, Salmonella typhimurium, Enterobacter aerogenes and Enterococcus faecalis*) **Standard:** Gentamycin and vancomycin (MIC < 0.05 mg/mL)	All bacterial strains showed MIC values superior to 1 mg/mL	[158]
Seed	Maceration (ethanol–water 95:05)	**Cannabinoids and polyphenols**	**Antibacterial** (against *Staphylococcus aureus)* **Standard:** Erythromycin (ID = 26.67 ± 1.15 mm)	Inhibition diameter = 1.00 mm	[155]
Seed	Maceration (ethanol–water 80:20)	**Polyphenols**	**Antibacterial** (against *Lactobacillus paracasei, Lactobacillus reuteri, Lactobacillus brevis, Lactobacillus plantarum, Bifidobacterium bifidum, Bifidobacterium longum and Bifidobacterium breve*) **Standard:** Gentamycin and vancomycin (MIC < 0.05 mg/mL)	All bacterial strains showed MIC values superior to 1 mg/mL	[158]
Seed	Maceration (ethanol–water 95:05)	**Terpenoid,** **phytocannabinoids and unsaturated fatty acids**	**Cytotoxicity**—SRB assay **Standard:** (not used)	Percentages of human skin fibroblast viability 66.12 ± 7.63% at 1 mg/mL	[155]
Seed	Maceration (ethanol–water 95:05)	**Terpenoids and flavonoids**	**Antiproliferative**—SRB assay **Standard:** Doxorubicin (IC_50_ = 0.0042 ± 0.0025 mg/mL)	On hepg2 cell lines: IC_50_ = 12.07 ± 1.18 mg/mL	[155]
Seed oil	Ultrasound-assisted extraction (methanol–water 80:20)	**Polyphenols**	**Antiproliferative** **Standard:** (not used)	Reduction in caco-2 and ht-29 cell viability to less than 40% from 150 mg/mL	[83]
Seed	Maceration (ethanol–water 95:05)	**Terpenoids and flavonoids**	**Anti-tyrosinase** **Standard:** Kojic acid (IC_50_ = 0.005 ± 0.004 mg/mL)	IC_50_ = 0.07 ± 0.06 mg/mL	[155]
Seed	Maceration (ethanol–water 75:25)	**Lignanamides**	**Acetyl choline esterase inhibition** **Standard:** Galantamine (IC_50_ = 2.76 μm)	**Cannabisin M, N and O** (IC_50_ = nd) **3,3′-demethyl-heliotropamide** (IC_50_ = 46.2 µm)	[156]
Leaf	Maceration (ethanol–water 95:05)	**Cannabinoids and** **polyphenols**	**Antioxidant**—DPPH **Standard:** Ascorbic acid (IC_50_ = 0.012 ± 0.002 mg/mL)	IC_50_ = 2.73 ± 0.422 mg/mL	[155]
Leaf	Maceration (ethanol–water 95:05)	**Cannabinoids and** **polyphenols**	**Antioxidant**—Metal ion chelating assay **Standard:** EDTA (CC_50_ = 0.15 ± 0.002 mg/mL)	CC_50_ = 0.93 ± 0.20 mg/mL	[155]
Leaf	Maceration (ethanol–water 95:05)	**Cannabinoids and** **polyphenols**	**Antioxidant**—Lipid peroxidation inhibition **Standard:** α-tocopherol (IPC_50_ = 0.045 ± 0.002 mg/mL)	IPC_50_ = 246.32 ± 69.38 mg/mL	[155]
Leaf	Ultrasound-assisted extraction (ethanol–water 80:20)	**Polyphenols and** **flavonoids**	**Antioxidant**—DPPH **Standard:** (not used)	Inhibition activity of 40% at 1000 µg/mL	[87]
Leaf	Maceration (acetone)	**ND**	**Antibacterial** **Standard:** Ampicillin (ID = 17.3 to 19.6 mm)	Inhibition diameters: *Escherichia coli* = 24.7 ± 1.5 mm *Staphylococcus aureus =* 19.6 ± 2.1 mm *Pseudomonas aeruginosa* = 19.0 ± 2.6 mm	[159]
Leaf	Maceration (chloroform)	**ND**	**Antibacterial** **Standard:** Ampicillin (ID = 17.3 to 19.6 mm)	Inhibition diameters: *Escherichia coli* = 23.0 ± 2.0 mm *Staphylococcus aureus =* 18.6 ± 2.08 mm *Pseudomonas aeruginosa* = 22.3 ± 1.52 mm	[159]
Leaf	Maceration (ethanol–water 60:40)	**ND**	**Antibacterial** **Standard:** Ampicillin (ID = 17.3 to 19.6 mm)	Inhibition diameters: *Escherichia coli* = 19.3 ± 1.2 mm *Staphylococcus aureus =* 18.6 ± 3.05 mm	[159]
Leaf	Maceration (ethanol–water 95:05)	**Polyphenols, flavones and polyholozides**	**Antibacterial** (against *Staphylococcus mutans)* **Standard:** Erythromycin (ID = 23 mm)	Inhibition diameter: 1.33 ± 0.58 mm	[155]
Leaf	Maceration (acetone)	**ND**	**Antifungal** **Standard:** (not used)	Inhibition diameters: *Aspergillus niger* = 21.3 ± 2 mm *Fusarium* spp. = 20 ± 2.64 mm	[159]
Leaf	Maceration (chloroform)	**ND**	**Antifungal** **Standard:** (not used)	Inhibition diameters: *Aspergillus niger* = 20.6 ± 1.5 mm *Fusarium* spp. = 18.3 ± 1.52 mm	[159]
Leaf	Maceration (ethanol–water 60:40)	**ND**	**Antifungal** **Standard:** (not used)	Inhibition diameters: *Aspergillus niger* = 23 ± 2 mm *Fusarium* spp. = 21.3 ± 3.21 mm	[159]
Leaf	Maceration (water)	**ND**	**Antifungal** **Standard:** (not used)	Inhibition diameters: *Aspergillus niger* = 21 ± 2.6 mm *Fusarium* spp. = 24.3 ± 3.51 mm	[159]
Leaf	Maceration (ethanol–water 95:05)	**Polyphenols, flavones and polyholozides**	**Cytotoxicity**—SRB assay **Standard:** (not used)	Percentages of human skin fibroblast viability 73.02 ± 3.57% at 1 mg/mL	[155]
Leaf	Ultrasound-assisted extraction (ethanol–water 80:20)	**Polyphenols and flavonoids**	**Cytotoxicity** **Standard:** (not used)	Viability of: HaCaT keratinocytes cells with 120% at 100 g/mL. BJ fibroblasts cells with 188% at 500 µg/mL.	[87]
Leaf	Maceration (ethanol–water 95:05)	**Polyphenols, flavones and polyholozides**	**Antiproliferative**—SRB assay **Standard:** Doxorubicin (IC_50_ = 0.0042 to 0.0274 mg/mL)	On hepg2 cell lines: IC_50_ = 13.17 ± 1.53mg/mL On kb cell lines: IC_50_ = 5.16 ± 1.66 mg/mL	[155]
Leaf	Maceration (ethanol–water 95:05)	**Terpenoids and flavonoids**	**Anti-tyrosinase** **Standard:** Kojic acid (IC_50_ = 0.005 ± 0.004 mg/mL)	IC_50_ = 0.049 ± 0.02 mg/mL	[155]
Leaf	Ultrasound-assisted extraction (ethanol–water 80:20)	**Cannabinoids**	**Anti-elastase** **Standard** (not used)	Inhibition value of 30% at 1000 µg/mL	[87]
Leaf	Ultrasound-assisted extraction (ethanol–water 80:20)	**Cannabinoids**	**Anti-collagenase** **Standard:** (not used)	Inhibition value of 80% at 1000 µg/mL	[87]
Leaf	Maceration (chloroform)	**Cannabinoids (THC, CBD and CBN)**	**Anticoagulant** **Standard:** (not used)	Inhibition values at 1 mg/mL: THC = 34.87% (IC50 = 1.79 mg/mL) CBN = 7.3% (IC50 = high value)	[160]
Flowers	Maceration (acetone–water 50:50)	**Polyphenols**	**Antifungal** (against *Aspergillus favus*)	Total inhibition at > 0.225 mg of DM/mL of medium	[161]
Flowers	Maceration (acetone–water 50:50)	**Polyphenols**	**Anti-aflatoxigenic** **Standard:** (not used)	Reduction in fungal growth by 36% at 7.2 mg DM/mL of medium	[161]
Inflorescences	Hydrodistillation (essential oil)	**Terpenes**	**Insecticidal** **Standard:** (not used)	Lethal concentration of LC_50_: *Anopheles stephensi* = 73.50 to 78.80 ppm for larvae. *Anopheles gambiae* = 20.13 to 67.19 ppm for pupae.	[162]
Inflorescences	Hydrodistillation (essential oil)	**Terpenes**	**Cytotoxicity** **Standard:** (not used)	IC50 values of: HaCaT cells = 2.23 ± 0.09 mg/mL Nhf a12 cells = 3.71 ± 0.2 mg/mL	[162]
Aerial parts	Hydrodistillation (essential oil + aromatic water)	**Terpenes and polyphenols**	**Antioxidant**—DPPH **Standard:** (not used)	Inhibition activity: Essential oil = 5.6 mg TE/g sample Aromatic water = 40 mg TE/g sample.	[94]
Aerial parts	Hydrodistillation (essential oil)	**Terpenes**	**Antioxidant**—DPPH **Standard:** Quercetin (IC_50_ = 1.1 ± 0.0 µg/mL)	IC_50_ = 1.6 ± 0.1 mg/mL	[163]
Aerial parts	Hydrodistillation (essential oil + aromatic water)	**Terpenes and polyphenols**	**Antioxidant**—FRAP **Standard:** (not used)	Inhibition activity: Essential oil = 57 mg TE/g sample Aromatic water = 83 mg TE/g sample.	[94]
Aerial parts	Hydrodistillation (essential oil + aromatic water)	**Terpenes and polyphenols**	**Antioxidant**—metal chelating activity **Standard:** (not used)	Inhibition activity: Essential oil = 19.3 mg EDTAE/g sample Aromatic water = 3.8 mg mg EDTAE/g DW	[94]
Aerial parts	Hydrodistillation (essential oil + aromatic water)	**Terpenes and polyphenols**	**Antioxidant**—phosphomolybdenum **Standard:** (not used)	Inhibition activity: Essential oil = 35.1 mmol TE/g of oil Aromatic water = 1.4 mmol TE/g DW	[94]
Aerial parts	Hydrodistillation (essential oil)	**Terpenes**	**Antioxidant**—reducing power assay **Standard:** Quercetin (EC_50_ = 2.3 ± 0.1 µg/mL)	EC_50_ = 1.8 ± 0.2 mg/mL	[163]
Aerial parts	Hydrodistillation (essential oil)	**Terpenes**	**Antioxidant**—β-carotene/linoleic acid **Standard:** Quercetin (EC_50_ = 0.9 ± 0.0 µg/mL)	EC_50_ = 0.9 ± 0.1 mg/mL	[163]
Aerial parts	Hydrodistillation (essential oil + aromatic water)	**Terpenes and polyphenols**	**Antioxidant**—CUPRAC **Standard:** (not used)	Inhibition activity: Essential oil = 141 mg TE/g sample Aromatic water = 109 mg TE/g sample.	[94]
Aerial parts	Hydrodistillation (essential oil)	**Terpenes and polyphenols**	**Antibacterial** (against *Helicobacter pylori* strains) **Standard:** Naringenin (MIC and MBC = 8–32 µg/mL)	MIC = 8–64 µg/mL MBC = 8–32 µg/mL	[94]
Aerial parts	Hydrodistillation (essential oil)	**Terpenes and polyphenols**	**Antibacterial** (against *Staphylococcus aureus*) **Standard:** (not used)	MIC = 8 mg/mL MBC = 16 mg/mL MBEC = 16–24 mg/mL	[94]
Aerial parts	Hydrodistillation (essential oil)	**Terpenes**	**Antibacterial** **Standard:** Ciprofloxacin (MIC = 0.015–1.00 mm)	*Micrococcus luteus* and *Staphylococcus aureus:* MIC = 4.7 mg/mL *Bacillus subtilis*, *Escherichia coli* and *Pseudomonas aeruginosa:* MIC = 1.2 mg/mL	[163]
Aerial parts	Hydrodistillation (essential oil)	**Terpenes**	**Antifungal** **Standard:** Fluconazole (MIC = 1.00 mm)	*Candida albicans*, *Candida glabrata*, *Candida krusei* and *Candida parapsilosis:* MIC = 9.5 mg/mL	[163]
Aerial parts	Hydrodistillation (essential oil)	**Terpenes and polyphenols**	**Antifungal** (against *Candida* spp. and *Malassezia* spp.) **Standard:** (not used)	MIC value > 12,460 µg/mL	[94]
Aerial parts	Hydrodistillation (essential oil + aromatic water)	**Terpenes and polyphenols**	**Cytotoxicity** **Standard:** Doxorubicin (IC_50_ = 3.1–23.3 µg/mL)	Inhibition activity of 50%: IC_50_ (*mda-mb-468)* = 53.0 µg/mL IC_50_ (caco-2) = 28.7 µg/mL IC_50_ (mz-cha-1) = 22.3 µg/mL	[94]
Aerial parts	Hydrodistillation (essential oil + aromatic water)	**Terpenes and polyphenols**	**Antidiabetic** **Standard:** (not used)	α-amylase inhibition: Essential oil = nd Aromatic water = 0.10 mmol ACAE/g extract α-glucosidase inhibition: Essential oil = 3.77 mmol ACAE/g oil Aromatic water = 0.17 mmol ACAE/g extract	[94]
Aerial parts	Hydrodistillation (essential oil + aromatic water)	**Terpenes and polyphenols**	**Acetyl- and butyryl-choline esterase inhibition** **Standard:** (not used)	Acetyl-choline esterase inhibition: Essential oil = nd Aromatic water = 2.56 mg GALAE/g extract Butyryl-choline esterase inhibition: Essential oil = 3.4 mg GALAE/g oil Aromatic water = 3.48 mg GALAE/g extract	[94]
Aerial parts	Hydrodistillation (essential oil + aromatic water)	**Terpenes and polyphenols**	**Lipase inhibition** **Standard:** (not used)	Inhibition activity: - essential oil = 70.14 mg OE/g oil - aromatic water = nd	[94]

IPC_50_: concentration providing 50% inhibition of lipid peroxidation/CC_50_: concentration providing 50% metal chelating activity. IC_50_: concentration of the sample that inhibits 50% expressed in mg/mL/ID: inhibition diameter (mm). EDTA: ethylenediaminetetraacetic acid./GAE: gallic acid equivalent./RE: rutin equivalent./CE: caffeic acid equivalent./TE: trolox equivalent. EDTAE: ethylenediaminetetraacetic acid equivalent./MIC: minimum inhibitory concentration./MBC: minimum bactericidal concentration. MCF-7: estrogen-dependent breast cancer cells./MDA-MB-468: triple-negative breast cancer cells./CACO-2: colorectal adenocarcinoma cells./MZ-CHA-1: cholangiocarcinoma cells.

## Data Availability

Data are contained within the article.

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
