# Peer review of "A Comprehensive Review on Cannabis sativa Ethnobotany, Phytochemistry, Molecular Docking and Biological Activities"

_plants, 2023, doi:10.3390/plants12061245_

Round 1

Reviewer 1 Report

In my opinion, the manuscript can be accepted for publication in this Journal after correcting few shortcomings:

- abbreviations THC, CBD, and CBG should be defined in line 300, abbreviation CBDV in line 327 where it appears for the first time, after that it is not necessary to repeat whole name of the compounds in the text, only abbreviations can be used

- style of citations in the text should be revised, it is not necessary mention all the authors of the publications, usual form of citations is “first author et. al”

- if the table is extended in several pages, table headings should be including in each page for easier reading of data in the tables

- it should be considered if the statement in lines 628-632 is accurate “The moderate activity was against Micrococcus luteus and Staphylococcus aureus, with an MIC of 4.7 mg/ml for both strains, whereas the highest inhibitory activity was observed against Escherichia coli, Pseudomonas aeruginosa and Bacillus subtilis, with an MIC of 1.2 mg/ml”. Concentration 4.7 mg/ml = 4.700 μg/ml is really high concentration. In case of antimicrobial assays, MIC values below 100 μg/ml for mixtures (extracts and essential oils) and, in the case of pure compounds, comparable with those of conventional antibiotics (0.001-16 μg/ml), can be considered as promising activity. The MIC of pure compounds higher than 100 μg/ml should strictly be evaluated as no active (for more information see DOI: 10.2174/0929867325666180831144344)

- the chapter “Dermocosmetic activities” is duplicit as chapters 3.4.8 and 3.5.10

Author Response

  • Abbreviations THC, CBD, and CBG should be defined in line 300, abbreviation CBDV in line 327 where it appears for the first time, after that it is not necessary to repeat whole name of the compounds in the text, only abbreviations can be used >> Done: The abbreviations had been added to the first mentions
  • Style of citations in the text should be revised, it is not necessary mention all the authors of the publications, usual form of citations is “first author et. al” >> Done: The references style had been adjusted, the new MDPI style had been inserted using Endnote.
  • if the table is extended in several pages, table headings should be including in each page for easier reading of data in the tables >> Done: The headings had been added to each table.
  • It should be considered if the statement in lines 628-632 is accurate “The moderate activity was against Micrococcus luteus and Staphylococcus aureus, with an MIC of 4.7 mg/ml for both strains, whereas the highest inhibitory activity was observed against Escherichia coli, Pseudomonas aeruginosa and Bacillus subtilis, with an MIC of 1.2 mg/ml”. Concentration 4.7 mg/ml = 4.700 μg/ml is really high concentration. In case of antimicrobial assays, MIC values below 100 μg/ml for mixtures (extracts and essential oils) and, in the case of pure compounds, comparable with those of conventional antibiotics (0.001-16 μg/ml), can be considered as promising activity. The MIC of pure compounds higher than 100 μg/ml should strictly be evaluated as no active >> Done: The interpretation of the antibacterial activity had been corrected as suggested.
  • the chapter “Dermocosmetic activities” is duplicit as chapters 3.4.8 and 3.5.10 >> Done: The duplicated section of the chapter had been deleted

Reviewer 2 Report

The current climatic and economic scenario pushes toward the use of sustainable resources to reduce our dependence on petrochemical and to minimize the impact on the environment. Plants are precious natural resources, because they can supply both phytochemicals and lignocellulosic biomass.

Cannabis sativa is an important herbaceous species originating from Central Asia, which has been used in folk medicine, food and as source of textile fiber since the dawn of times. Equally highly interested in this plant are the pharmaceutical and construction sectors, since its metabolites show potent bioactivities on human health and its outer and inner stem tissues can be used to make bioplastics and concrete-like material respectively.

Cannabis is one of the most popular recreational drugs; worldwide, an estimated 178 million people aged 15 to 64 years used cannabis at least once in 2012. Medical cannabis refers to use of cannabis or cannabinoids as medical therapy to treat disease or alleviate symptoms.

This shows the great versatility of this fiber crop and encourages future studies focused on Cannabis.

The aim of this review is to describe the traditional uses, chemical composition and biological activities of different parts of this plant as well as the molecular docking studies. Information was collected from electronic databases namely SciFinder, Science Direct, PubMed and Web of Science.

This extensive review on Cannabis sativa covers a broad range of topics:

·        Generalities about Cannabis (plant nomenclature and synonyms, description and botanical aspect, geographic distribution and history);

·        Traditional uses of Cannabis sativa;

·        Chemical composition of Cannabis sativa;

·        Molecular docking studies of Cannbis (pesticidal activity, antimalarial and anti-leishmania activity, antiviral, anti-inflammatory activity, anti-cancer activity, antiepileptic, neuroprotective activity, dermatocosmetic activity);

·        Biological activities of Cannabis (antioxidant activity, antimicrobial activity, insecticidal activity, anticoagulant activity, antidiabetic activity, anticancer activity);

·        Drugs based on Cannabis

The amount of scientific information and its systematization is remarkable. The presentation of the traditional medical uses and the chemical composition of each plant part (tables 1 and 2) catch the reader`s attention from the first pages.

The data is well synthesized, yet the authors have also managed to provide details concerning extraction methods, bioactive metabolites, biological activities etc. The objectives of this study have certainly been met.

Accept in present form.

Author Response

No remarks

Reviewer 3 Report

The article under review addresses aspects of Cannabis sativa phytochemistry, molecular docking and bioactivity. It offers a good perspective on the botany, composition of various extracts from different plant parts, and knowledge derived from docking studies. However, some important points should be addressed:

- The article presents traditional uses of the Cannabis plant, but is mostly based books and other review papers, and very few original articles.  

- General aspects, Description and botanical aspect are chapters that sould not be included in “Results and Discussion” section, but in the introduction.

- In the Introduction, you mention a fourth subspecies, ssp. afghanica (line 43), which does not appear in the General aspects section (line 106). Are there 3 or 4 subspecies?

- Humulus and Pteroceltis mus be written with first letter capitalized (line 103).

- Bibliographical information should be updated. There are surely more recent references besides Fournier et al. 1981 that show  that the THC content of a Cannabis sativa single species depends on the growing climate of the plant”. While older references (1975, ‘76) do bring a historical perspective, new original articles on the taxonomy of Cannabis should imperatively be cited.

- Line 110: all abbreviations should be explained the first time they are used (THC etc)

- The main purpose of a review article is to summarize data from original articles. However, the paper under peer review is much based on other review articles (most references from 1-73). Chapters that based on references from other reviews and books should not be included under Results and Discussions. New research results on Cannabis extracts or pure metabolites are mostly highlighted from reference 74.

- To cite references in text, for ex. Babiker, Uslu, Al Juhaimi, Ahmed, Ghafoor, Özcan and Almusallam [81] : cite as Babiker et al (first author et al)

Author Response

  • General aspects, Description and botanical aspect are chapters that should not be included in “Results and Discussion” section, but in the introduction >> Done: The generalities had been moved before the methodology and results/ discussion sections
  • In the Introduction, you mention a fourth subspecies, ssp. afghanica (line 43), which does not appear in the General aspects section (line 106). Are there 3 or 4 subspecies? >> Done: In fact there is still controversy on the number of subspecies, we have cited the four subspecies reported on the literature, we rectified the line 106 as remarked.
  • Humulus and Pteroceltis must be written with first letter capitalized (line 103) >> Done: The genus names had been capitalized
  • Bibliographical information should be updated. There are surely more recent references besides Fournier et al. 1981 that show “that the THC content of a Cannabis sativa single species depends on the growing climate of the plant”. While older references (1975, ‘76) do bring a historical perspective, new original articles on the taxonomy of Cannabis should imperatively be cited. >> Done: The citations had been updated
  • Line 110: all abbreviations should be explained the first time they are used (THC etc)>> Done: The abbreviations had been explained in their first mention.
  • The main purpose of a review article is to summarize data from original articles. However, the paper under peer review is much based on other review articles (most references from 1-73). Chapters that based on references from other reviews and books should not be included under Results and Discussions. New research results on Cannabis extracts or pure metabolites are mostly highlighted from reference 74 >> Done: The generalities about Cannabis had been moved to the introduction part (before methodology and results), the use of reviews on this part will be kept. However we changed the references used on the traditional part (we replaced the reviews references, with citations from original papers).
  • To cite references in text, for ex. Babiker, Uslu, Al Juhaimi, Ahmed, Ghafoor, Özcan and Almusallam [81] : cite as Babiker et al (first author et al) >> Done: The references style had been adjusted, the new MDPI style had been inserted using Endnote.

Round 2

Reviewer 1 Report

Dear authors, I appreciate your effort to improve quality of your manuscript, however I recommend other revisions to make your manuscript acceptable for publishing.

- First time full name of “THC” (line 99) should be included.

- Writing of abbreviations should be checked within the manuscript, if the abbreviation is explained for the first time, it is not necessary to repeat the full name when it is mention again

- Writing of subscripts should be checked within the manuscripts (e.g. IC50).

- In Table 4, names of bacteria should start with capital. It is recommended to include names of the assays used in case of antimicrobial activity.

- It is recommended to unified units within the manuscript for easier comparison of the results (e.g. units mg/ml and μg/ml in chapters of “Biological activities of Cannabis sativa” section).

Author Response

  • Point 1:  First time full name of “THC” (line 99) should be included.
  • Response 1: The abbreviations had been added to the first mentions
  • Point 2: Writing of abbreviations should be checked within the manuscript, if the abbreviation is explained for the first time, it is not necessary to repeat the full name when it is mention again
  • Response 2: The explanation of abbreviations had been checked and corrected in the whole manuscript
  • Point 3: Writing of subscripts should be checked within the manuscripts (e.g. IC50).
  • Response 3: Checked and rectified
  • Point 4: In Table 4, names of bacteria should start with capital. It is recommended to include names of the assays used in case of antimicrobial activity.
  • Response 4: The bacteria and other living organisms names had been checked and rectified in the tables and full text
  • Point 5: It is recommended to unified units within the manuscript for easier comparison of the results (e.g. units mg/ml and μg/ml in chapters of “Biological activities of Cannabis sativa” section).
  • Response 5: In a first time we applied this modification however it gave us too much long values and too much small values, the 10x form can be applied but it is only applicable for the microbiological part, when the pharmacological and enzymatic assays have to be written in full form, for structural reasons we prefer to keep the units as they are. However we have already shown the difference between the values in the discussion part.

Reviewer 3 Report

The authors updated their manuscript sufficiently to warrant publication.

Author Response

No comments to answer